# A Resolution-Agnostic Geometric Transformer for Chromosome Modeling Using Inertial Frame

**Yize Zhou**[1,2*]**, Haorui Li**[1,2]**, Shengchao Liu**[1,2]
[1]Wave Intelligence Lab
[2]Department of Computer Science and Engineering, The Chinese University of Hong Kong

## Abstract

Chromosomes are the carriers of genetic information. Further understanding their 3D structure can help reveal gene-regulatory mechanisms and cellular functions. However, high-resolution 3D structures are often missing due to the high cost and inherent noise of experimental screening. A standard pipeline for reconstructing the chromosome 3D structure first applies the single-cell Hi-C high-throughput screening method to measure pairwise interactions between DNA fragments at different resolutions; then it adopts computational methods to reconstruct the 3D structures from these contacts. These include traditional numerical methods and deep learning models, which struggle with limited model expressiveness and poor generalization across resolutions. To handle this issue, we propose InertialGenome, a novel transformer-based framework for robust and resolution-agnostic chromosome reconstruction. InertialGenome first adopts the inertial frame for the pose canonicalization. Then, based on such an invariant pose, it proposes a Transformer with geometry-aware positional encoding, leveraging Nyström estimation. To verify the effectiveness of InertialGenome, we conduct experiments on two single-cell 3D reconstruction datasets with four resolutions, reaching superior performance over all four computational baselines. Additionally, we observe that the 3D structure reconstructed by InertialGenome is more in line with the results of real experimental results on two functional verification tasks. Finally, we leverage InertialGenome for cross-resolution transfer learning, yielding up to a 5% improvement from low to high resolution. The source code is available at https://github.com/yize1203/InertialGenome.

## 1 Introduction

The genome encodes the complete set of genetic information within an organism, stored as a full DNA sequence. This information is packaged into chromosomes, which serve as the carriers of genetic material and compact the DNA into three-dimensional structures. These chromosomes adopt complex 3D conformations that play essential roles in gene regulation, cell differentiation, and disease progression (Lieberman-Aiden et al., 2009; Dixon et al., 2012; Rao et al., 2014). Importantly, such an organization cannot be inferred from the linear DNA sequence alone (Consortium et al., 2024).

Over the past two decades, diverse experimental techniques have emerged to probe the 3D conformations. Early approaches such as 3C(Dekker et al., 2002), 4C(Simonis et al., 2006), and 5C(Dostie et al., 2006) enable targeted interrogation of chromatin interactions at specific loci. More recent genome-wide methods, including ChIA-PET(Fullwood et al., 2009), SPRITE(Quinodoz et al., 2022), GAM(Beagrie et al., 2017), and Hi-C(Belton et al., 2012), provide comprehensive maps of spatial chromatin contacts across the entire genome. Among them, high-throughput chromosome conformation capture (Hi-C) has become the gold standard for 3D modeling because it enables the systematic profiling of interactions between all genomic loci simultaneously. In Hi-C, the genome is partitioned into consecutive, non-overlapping segments called *bins*, with their length determined

---

[*]Work done during an internship at Wave Intelligence Lab.

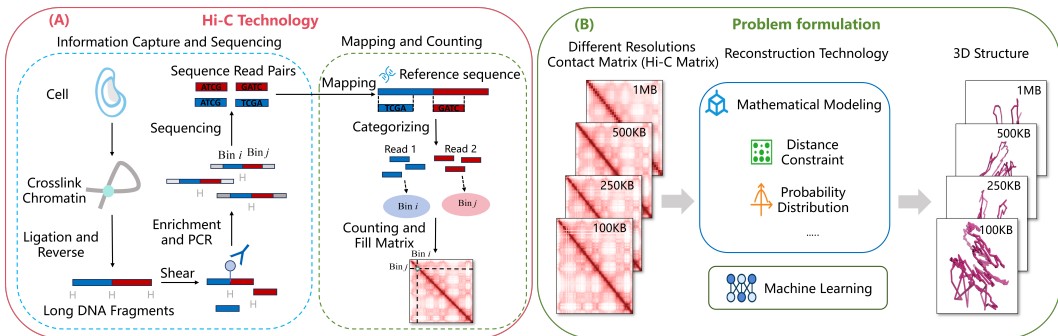

Figure 1: Overview of chromosome 3D reconstruction using Hi-C technology. (A): Experimental implementation of Hi-C for obtaining contact matrix information. (B): Computational pipeline for 3D structure reconstruction via mathematical modeling or machine learning based on the Hi-C contact matrix.

by the chosen *resolution* (*e.g.*, 1 kb, 10 kb, or 250 kb per bin, with higher resolution corresponding to shorter segments). As a result, Hi-C produces a contact matrix that records, for every pair of bins, the frequency of spatial contacts between their underlying DNA segments, as illustrated in Figure 1A. Hi-C is chosen for our task because maps at different resolutions offer complementary structural information: high-resolution maps capture fine local structures but are often sparse and noisy, while low-resolution maps are denser and more robust, reflecting global organization. Driven by this observation, we focus on this *cross-resolution* task by using low-resolution maps as structural priors to guide high-resolution 3D genome reconstruction.

Following the Hi-C matrix, the next step is to computationally reconstruct the 3D conformations (Trieu & Cheng, 2017; Oluwadare et al., 2018; Wang & Cheng, 2025). Traditional numerical methods based on distance geometry, such as ChromSDE (Zhang et al., 2013), miniMDS (Rieber & Mahony, 2017), and 3DMAX (Oluwadare et al., 2018), attempt to find optimal coordinates by minimizing distance-based objective functions. However, these approaches are computationally intensive because the underlying non-convex optimization involves high-dimensional search spaces and quadratic complexity in distance matrix operations. Consequently, they often struggle to converge on high-resolution genomic data and are largely limited to low-resolution modeling due to data sparsity. To address these issues, deep learning offers an efficient, data-driven alternative. For instance, HiC-GNN (Hovenga et al., 2023) leverages node embeddings and graph neural networks to predict 3D conformation directly from Hi-C contact graphs; HiCEGNN (Wang & Cheng, 2025) further incorporates E(3)-equivariance as a model constraint. However, these approaches share some key limitations: they rely solely on Hi-C contacts without incorporating explicit geometric priors (e.g., chromatin principal axes or directional chain structure). Moreover, the strong symmetry constraints of structures such as HiCEGNN limit model expression ability, making it difficult to process asymmetric structures (such as anchored loops).

**Our Contributions.** To tackle this problem, we propose InertialGenome, a novel Transformer-based framework for 3D chromosome reconstruction. InertialGenome has two main components. (1) InertialGenome performs pose canonicalization. It aligns each chromosome to its inertial frame—a coordinate system defined by the principal axes of its inertia tensor. This tensor is computed from the 3D point cloud of the chromosome. The alignment removes arbitrary rotations and translations, resulting in a pose-invariant representation. (2) Based on this invariant frame, InertialGenome employs a geometry-aware positional encoding into the Transformer architecture. The core idea is to project the bin-wise positions into an imaginary space and use the Nyström method to estimate the pairwise distance via inner products. Specifically, Nyström enables efficient low-rank estimation of the radial basis function (RBF) kernel over 3D coordinates, capturing long-range structural dependencies without computing the full distance matrix. To verify the effectiveness of InertialGenome, we conduct experiments on two single-cell 3D chromosome reconstruction datasets at four resolutions, where it consistently outperforms four baselines in both two structure metrics and two functional validation tasks. Additionally, InertialGenome excels at cross-resolution transfer tasks, reaching up to a 5% performance improvement.

**Related Work.** Existing methods for 3D chromosome reconstruction fall into three categories (see Appendix A): **Distance-based methods** (*e.g.*, 3DMAX (Oluwadare et al., 2018), LorDG (Trieu & Cheng, 2017)) convert contacts to distance constraints; **Probabilistic approaches** (*e.g.*, BACH (Hu et al., 2013), PASTIS (Varoquaux et al., 2014)) model contact matrices as observations from spatial distributions; **Deep learning methods** (*e.g.*, HiC-GNN (Hovenga et al., 2023), HiCEGNN (Wang & Cheng, 2025)) map interactions to 3D structures via neural networks. However, these methods have some limitations: they rely on simplistic modeling of contact matrices as the sole input, lacking deeper structural interpretation, and their model expression ability may be constrained. Inspired by the success of InertialAR (Li et al., 2025), which leverages inertial frames and geometric encoding for invariant molecule generation, we propose InertialGenome. Our framework extends these geometric principles to chromosome modeling, integrating inertial frame canonicalization and geometry-aware positional encoding to significantly improve robustness across resolutions.

## 2 Preliminaries

The 3D chromosome reconstruction methods typically take the Hi-C contact matrix as input. The resolution of a Hi-C contact matrix reflects the length of chromosome segments, with higher resolution corresponding to shorter segments. For example, at 1 kb resolution, each 1,000 base pairs forms a segment. When applied to the human genome with standard masking of unmappable regions, this yields approximately 248,947 bins. This underscores the sharp scale differences across resolutions, with high-resolution 3D reconstruction being far more computationally demanding and challenging.

The reconstruction process consists of two key steps, as shown in Figure 1B. First, the contact frequencies $IF_{ij}$ are converted to spatial distances $D_{ij}$ based on the inverse relationship between distance and contact frequency (Pombo & Nicodemi, 2014; Barbieri et al., 2012), expressed as $D_{ij} = IF_{ij}^{-\gamma}$ where $\gamma \in [0.1, 0.2, \ldots, 2]$. Second, the 3D coordinates $(x_i, y_i, z_i)$ are inferred from these distances using either numerical methods or deep learning methods. In the following paper, we denote $t_i$ as the index of the $i$-th bin within the chromosome. Its canonicalized 3D coordinates are represented as $\mathbf{s}_i = (\mathbf{s}_{x_i}, \mathbf{s}_{y_i}, \mathbf{s}_{z_i}) \in \mathbb{R}^3$, which are pose-normalized via alignment to the chromosome's principal axes, as detailed in Section 3.1.

**Problem formulation.** In this work, we are interested in solving the 3D chromosome reconstruction task. Following the existing paradigm (Hovenga et al., 2023; Wang & Cheng, 2025), the first step is to apply a numerical method to generate an initial position $\mathbf{C}^* \triangleq \{(x_i^*, y_i^*, z_i^*)\}_{i=1}^N$, where $N$ is the number of bins (*e.g.*, nodes), from Hi-C contact matrix. Then, our model takes these initial 3D coordinates $\mathbf{C}^*$ as the input, and the output is an accurately reconstructured 3D coordinates $\hat{\mathbf{C}} \triangleq \{(\hat{x}_i, \hat{y}_i, \hat{z}_i)\}_{i=1}^N$. Rigorously, we are solving the 3D chromosome reconstruction task as $\hat{\mathbf{C}} = f(\mathbf{C}^*)$.

## 3 Method InertialGenome

In this section, we introduce InertialGenome, a novel Transformer-based framework for robust and resolution-agnostic chromosome reconstruction. It consists of three key components: inertial frame canonicalization, geometry-aware positional encoding, and structure-aware fusion. Figure 2 shows the whole architecture of InertialGenome.

### 3.1 Inertial Frame Canonicalization

To achieve pose-invariant representation of 3D chromosome structures, we implement an inertial frame canonicalization method, with the following steps: (1) **Centroid translation**: $\bar{c} = \frac{1}{N} \sum_{i=1}^N \mathbf{c}_i$, where $\mathbf{c}_i \in \mathbf{C}^*$. It will adjust position relative to the center $\mathbf{c}_i' = \mathbf{c}_i - \bar{c}$. (2) **Inertia tensor computation**: We estimate the normalized inertia tensor as $\hat{\mathbf{I}} = \frac{1}{N} \sum_{i=1}^N \left( \|\mathbf{c}_i'\|^2 \mathbf{I}_3 - \mathbf{c}_i'(\mathbf{c}_i')^T \right)$, where $\mathbf{I}_3$ is the $3 \times 3$ identity matrix. (3) **Principal axes alignment**: We perform eigendecomposition of the inertia tensor $\hat{\mathbf{I}} = L\Lambda L^T$, where $\Lambda = \mathrm{diag}(\lambda_x, \lambda_y, \lambda_z)$ contains the eigenvalues with $\lambda_x \geq \lambda_y \geq \lambda_z$. The columns of $L$ are the corresponding orthonormal eigenvectors $\mathbf{l}_x, \mathbf{l}_y, \mathbf{l}_z$, which define the principal axes in descending eigenvalue order. (4) **Chirality correction**: We select the farthest point $\mathbf{c}_{\max} = \arg\max_i \|\mathbf{c}_i'\|$ and map it into the principal-axis frame: $\mathbf{p} = L^\top \mathbf{c}_{\max}$,

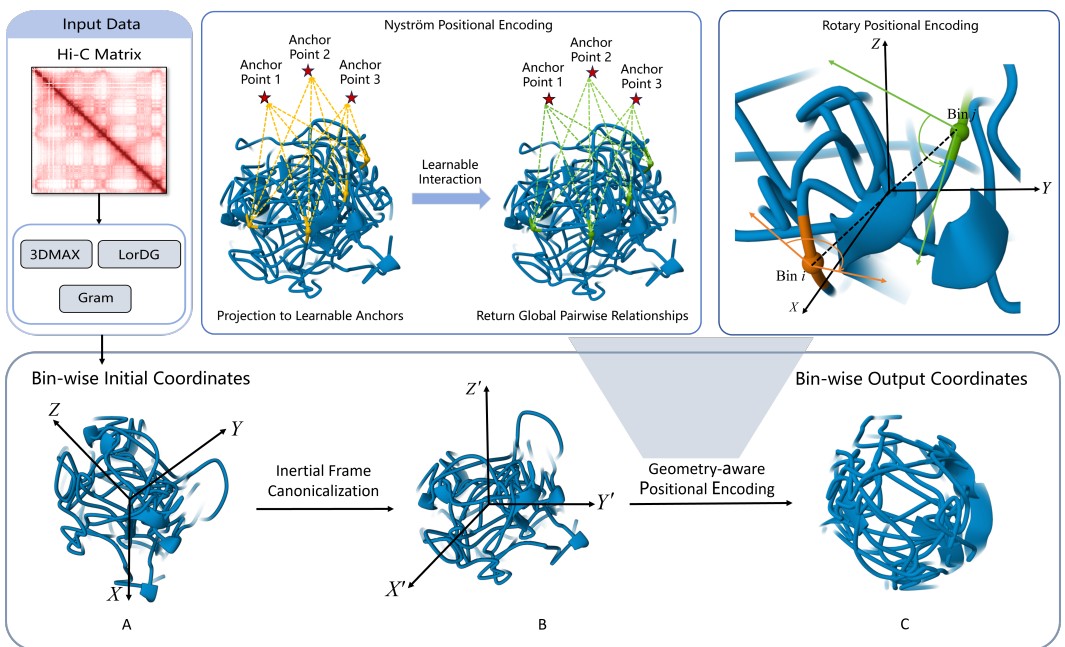

Figure 2: The architecture of InertialGenome. The framework takes 3D chromosome coordinates, initialized from Hi-C matrices using numerical methods, as input Data. (A) The model first performs inertial frame canonicalization to transform raw coordinates into a pose-invariant representation by aligning them with the chromosome's principal axes $(X', Y', Z')$. (B) These canonicalized coordinates are processed by the Geometry-aware Positional Encoding mechanism (highlighted by the shaded projection). This component integrates Nyström Positional Encoding for efficient global relationship modeling via learnable anchors, and Rotary Positional Encoding for pairwise distance awareness. (C) Finally, the framework outputs Corrected Coordinates, yielding stable and refined 3D chromosome structures.

where $\mathbf{p} = (p_x, p_y, p_z)$ is the coordinate of the farthest point expressed in the principal-axis basis. We then adjust the first two axes as $\mathbf{l}_x \leftarrow \text{sign}(p_x)\mathbf{l}_x$, $\mathbf{l}_y \leftarrow \text{sign}(p_y)\mathbf{l}_y$, and enforce a right-handed system by $\mathbf{l}_z = \mathbf{l}_x \times \mathbf{l}_y$. Then, the canonical transform is $R = [\mathbf{l}_x, \mathbf{l}_y, \mathbf{l}_z]^T$. Pose-invariant coordinates are obtained as $\mathbf{s}_i = R\mathbf{c}'_i$, where $\mathbf{s}_i \in \mathbf{S}$, $\mathbf{S} \triangleq \{\mathbf{s}_{x_i}, \mathbf{s}_{y_i}, \mathbf{s}_{z_i}\}_{i=1}^N$. Although chromosome structures naturally occupy 3D space, reconstruction algorithms may yield degenerate cases (e.g., nearly coplanar nodes). We discard such samples before training. After applying the canonical transform, each coordinate is represented as

$$\mathbf{s}_i = R\,\mathbf{c}'_i, \quad \mathbf{S} \triangleq \{\mathbf{s}_i\}_{i=1}^N \subset \mathbb{R}^3. \tag{1}$$

## 3.2 GEOMETRY-AWARE POSITIONAL ENCODING

In 3D chromosome structure modeling, we need a position embedding method that can maintain both absolute position information and the relative distance when calculating the inner product, *e.g.*, $R_{\mathbf{s}_{x_1}, \mathbf{s}_{y_1}, \mathbf{s}_{z_1}}^T R_{\mathbf{s}_{x_2}, \mathbf{s}_{y_2}, \mathbf{s}_{z_2}} = R_{\mathbf{s}_{x_1} - \mathbf{s}_{x_2}, \mathbf{s}_{y_1} - \mathbf{s}_{y_2}, \mathbf{s}_{z_1} - \mathbf{s}_{z_2}}$.

**(1) Geometric Position Encoding with RoPE.** Inspired by Su et al. (2024), we extend Rotary Position Encoding (RoPE) to 3D Euclidean space. To achieve rotation-equivariant attention while maintaining pairwise distance awareness, we decompose 3D spatial encoding into three independent 2D rotary subspaces corresponding to the $(x, y)$, $(y, z)$, and $(z, x)$ planes. This yields a 6-dimensional geometric embedding structure, organized into three pairs, each pair encoding angular information along one coordinate axis.

The 3D rotary position encoding is applied to the projected query and key vectors. Let $\boldsymbol{q}^{\text{raw}}, \boldsymbol{k}^{\text{raw}} \in \mathbb{R}^d$ denote the query and key vectors after linear projection but before positional encoding, associated

with 3D coordinate $\mathbf{s}_i = (s_{x_i}, s_{y_i}, s_{z_i})$. We define the rotation operator $R_{\mathbf{s}_x, \mathbf{s}_y, \mathbf{s}_z}$ as:

$$
R_{\mathbf{s}_x, \mathbf{s}_y, \mathbf{s}_z} \, \boldsymbol{q}^{\mathrm{raw}} = \begin{bmatrix} q_0^{\mathrm{raw}} \\ q_1^{\mathrm{raw}} \\ q_2^{\mathrm{raw}} \\ q_3^{\mathrm{raw}} \\ q_4^{\mathrm{raw}} \\ q_5^{\mathrm{raw}} \end{bmatrix} \odot \begin{bmatrix} \cos(s_x\theta_0) \\ \cos(s_x\theta_0) \\ \cos(s_y\theta_0) \\ \cos(s_y\theta_0) \\ \cos(s_z\theta_0) \\ \cos(s_z\theta_0) \end{bmatrix} + \begin{bmatrix} -q_1^{\mathrm{raw}} \\ q_0^{\mathrm{raw}} \\ -q_3^{\mathrm{raw}} \\ q_2^{\mathrm{raw}} \\ -q_5^{\mathrm{raw}} \\ q_4^{\mathrm{raw}} \end{bmatrix} \odot \begin{bmatrix} \sin(s_x\theta_0) \\ \sin(s_x\theta_0) \\ \sin(s_y\theta_0) \\ \sin(s_y\theta_0) \\ \sin(s_z\theta_0) \\ \sin(s_z\theta_0) \end{bmatrix}, \tag{2}
$$

where $\odot$ denotes element-wise multiplication. The final geometry-aware query and key are given by $\boldsymbol{q} = R_{\mathbf{s}_i} \boldsymbol{q}^{\mathrm{raw}}$ and $\boldsymbol{k} = R_{\mathbf{s}_i} \boldsymbol{k}^{\mathrm{raw}}$, respectively. This formulation ensures that the inner product satisfies the relative positional property: $(R_{\mathbf{s}_1} \boldsymbol{q}^{\mathrm{raw}})^\top (R_{\mathbf{s}_2} \boldsymbol{k}^{\mathrm{raw}}) = (\boldsymbol{q}^{\mathrm{raw}})^\top R_{\mathbf{s}_1 - \mathbf{s}_2} \boldsymbol{k}^{\mathrm{raw}}$, which encodes 3D spatial relationships directly into attention scores. A detailed derivation is provided in Appendix B.

We implement three RoPE modes that differ in how the input embedding is processed before applying 3D rotary position encoding:

- **Selective**: The input embedding $\boldsymbol{x}_i \in \mathbb{R}^d$ is split into two halves. The first half (spatial part) is linearly projected and then transformed by 3D RoPE; the second half (feature part) is kept unchanged and directly concatenated.
- **Separate**: The two halves are independently linearly projected, but only the first half is connected through 3D RoPE.
- **Full**: The entire embedding is linearly projected as a vector and fully transformed by 3D RoPE.

**Input representation.** Each token $t_i$ corresponds to a genomic bin with known 3D coordinates $\mathbf{s}_i = (s_{x_i}, s_{y_i}, s_{z_i}) \in \mathbb{R}^3$. The total number of bins for a chromosome at a given resolution defines the vocabulary size. We map each bin ID $t_i \in \{0, \ldots, \texttt{vocab\_size} - 1\}$ to a learnable semantic embedding $\mathbf{E}_{\mathrm{token}}(t_i) \in \mathbb{R}^{d_t}$ via a matrix $\mathbf{W}_{\mathrm{tok}} \in \mathbb{R}^{\texttt{vocab\_size} \times d_t}$.

The initial token representation is formed by concatenating this semantic embedding with the raw spatial coordinates:

$$
\boldsymbol{x}_i = \big[\, \mathbf{E}_{\mathrm{token}}(t_i); \, \mathbf{s}_i \,\big] \in \mathbb{R}^d, \tag{3}
$$

Let $\boldsymbol{x}_i^{(s)}$ and $\boldsymbol{x}_i^{(f)}$ denote the spatial and feature halves of $\boldsymbol{x}_i$, each of dimension $d/2$. The 3D rotary position embedding is applied as follows:

$$
\text{3D-RoPE}(\boldsymbol{x}_i) = \begin{cases} \big[\, R_{\mathbf{s}_i}\big(W^{\mathrm{rope}} \boldsymbol{x}_i^{(s)}\big); \, \boldsymbol{x}_i^{(f)} \,\big] & \text{(Selective)} \\[2mm] \big[\, R_{\mathbf{s}_i}\big(W^{\mathrm{rope}} \boldsymbol{x}_i^{(s)}\big); \, W^{\mathrm{feat}} \boldsymbol{x}_i^{(f)} \,\big] & \text{(Separate)} \\[2mm] R_{\mathbf{s}_i}\big(W \boldsymbol{x}_i\big) & \text{(Full)} \end{cases}, \tag{4}
$$

where $W^{\mathrm{rope}}, W^{\mathrm{feat}}, W$ are learnable linear projections, and $R_{\mathbf{s}_i} = R_{s_{x_i}, s_{y_i}, s_{z_i}}$ is the 6D rotary transformation defined in Equation (2).

**(2) Nyström Positional Encoding for Structure Tokenization.** While 3D-RoPE in Equation (2) effectively encodes absolute and relative spatial positions, its axis-wise rotation mechanism is inherently limited in modeling global pairwise distance relationships, such as long-range structural dependencies or non-local geometric patterns. To address this, we incorporate a Nyström-based feature encoder (Williams & Seeger, 2000; Yang et al., 2012) that explicitly captures low-rank approximations of the radial basis function (RBF) kernel over 3D coordinates.

Formally, let $\mathbf{s}_i \in \mathbb{R}^3$ denote the canonicalized coordinate of token $i$. We define an RBF kernel between any two points as:

$$
\kappa_g(\mathbf{s}_i, \mathbf{s}_j) = \exp\left( -\frac{\|\mathbf{s}_i - \mathbf{s}_j\|^2}{2\sigma_g^2} \right),
$$

where $\sigma_g > 0$ is the bandwidth of the $g$-th Gaussian kernel, and we consider a set of $G$ scales $\{\sigma_g\}_{g=1}^G$. The Nyström method proceeds as follows:

**Step 1: Anchor point selection.** We fix a set of $m$ anchor points $\{\mathbf{u}_k\}_{k=1}^m \subset \mathbb{R}^3$, sampled uniformly from the 3D space. For each scale $\sigma_g$, we construct the anchor–anchor Gram matrix $A_g \in \mathbb{R}^{m \times m}$ with entries:

$$
[A_g]_{k\ell} = \kappa_g(\mathbf{u}_k, \mathbf{u}_\ell) = \exp\left( -\frac{\|\mathbf{u}_k - \mathbf{u}_\ell\|^2}{2\sigma_g^2} \right).
$$

To ensure numerical stability, we compute the Cholesky decomposition $A_g = O_g O_g^\top$, where $O_g \in \mathbb{R}^{m \times m}$ is a lower-triangular matrix, and precompute $O_g^{-\top}$ for later use.

**Step 2: Token–anchor similarity.** For each token coordinate $\mathbf{s}_i$, we compute its RBF similarities to all anchors under scale $\sigma_g$:

$$V_{g,i} = \big[\kappa_g(\mathbf{s}_i, \mathbf{u}_1),\ \kappa_g(\mathbf{s}_i, \mathbf{u}_2),\ \ldots,\ \kappa_g(\mathbf{s}_i, \mathbf{u}_m)\big] \in \mathbb{R}^m.$$

**Step 3: Nyström projection.** We project $V_{g,i}$ using the precomputed inverse factor:

$$\tilde{k}_{g,i} = V_{g,i}\, O_g^{-\top} \in \mathbb{R}^m.$$

This yields a low-rank approximation of the full kernel embedding.

**Step 4: Multi-scale fusion and compression.** We concatenate the projected features across all scales:

$$\tilde{k}_i = \big[\tilde{k}_{1,i};\ \tilde{k}_{2,i};\ \ldots;\ \tilde{k}_{G,i}\big] \in \mathbb{R}^{Gm},$$

and apply a learnable linear projection $f_\theta : \mathbb{R}^{Gm} \to \mathbb{R}^m$ to obtain the final Nyström structure embedding:

$$\mathbf{E}_{\text{nyström}}(\mathbf{s}_i) = f_\theta\big(\tilde{k}_i\big). \tag{5}$$

This embedding encodes multi-scale, low-rank geometric information about $\mathbf{s}_i$ and is fused with token and positional representations in the subsequent transformer layers.

## 3.3 STRUCTURE-AWARE FUSION

We fuse the geometry-aware positional encoding and Nyström structure features into a unified Transformer input.

**Input Representation.** For each token $i$, we construct the initial embedding by concatenating three geometric components: (1) the base position embedding $\boldsymbol{x}_i$, (2) the normalized canonical coordinate $\frac{\mathbf{s}_i}{\|\mathbf{s}_i\|}$, which encodes directional information, and (3) the Nyström structure embedding $\mathbf{E}_{\text{nyström}}(\mathbf{s}_i)$. This yields:

$$\mathbf{h}_i^0 = \text{Concat}\Big(\boldsymbol{x}_i,\ \frac{\mathbf{s}_i}{\|\mathbf{s}_i\|},\ \mathbf{E}_{\text{nyström}}(\mathbf{s}_i)\Big) \in \mathbb{R}^{d_{\text{in}}}. \tag{6}$$

**Position-Augmented Transformer Input.** We then add the geometry-aware positional encoding 3D-RoPE$(\boldsymbol{x}_i)$ to inject relative spatial context, followed by dropout:

$$\mathbf{H}_0 = \text{Dropout}\big(\mathbf{h}^0 + \text{3D-RoPE}(\boldsymbol{x}_i)\big). \tag{7}$$

**Transformer Backbone.** The sequence $\mathbf{H}_0$ is processed by $L$ stacked Transformer layers:

$$\mathbf{H}_{l+1} = \text{TransformerLayer}_l(\mathbf{H}_l), \quad l = 0, \ldots, L-1. \tag{8}$$

The final representation $\mathbf{H}_L$ aggregates multi-scale geometric and structural information, and is used to predict the 3D coordinates of chromosome bins.

## 3.4 LEARNING OBJECTIVE

The overall training objective is a hybrid loss that combines a structural-preserving term and a value-weighted regression term:

$$\mathcal{L}_{\text{total}} = \alpha\, \mathcal{L}_{\text{struct}} + \beta\, \mathcal{L}_{\text{weighted\_mse}}, \qquad \beta = 1 - \alpha, \tag{9}$$

where $\alpha \in [0,1]$ balances the two components.

**Structural-learning loss.** Let $D = \{D_{ij}\}$ denote the input pairwise distance representation derived from contact frequencies, and let $\hat{\mathbf{C}} = \{\hat{\mathbf{s}}_i\}_{i=1}^N$ be the predicted 3D coordinates where $\hat{\mathbf{s}}_i \in \mathbb{R}^3$. To capture the local topology, we define neighborhood selection probabilities for each bin $i$ in both the input and output (reconstructed) spaces:

$$p_{j|i} = \frac{\exp(-D_{ij})}{\sum_{k \neq i} \exp(-D_{ik})}, \qquad q_{j|i} = \frac{\exp(-\|\hat{\mathbf{s}}_i - \hat{\mathbf{s}}_j\|^2)}{\sum_{k \neq i} \exp(-\|\hat{\mathbf{s}}_i - \hat{\mathbf{s}}_k\|^2)}. \tag{10}$$

To align the neighborhood structures between these spaces, we employ a bidirectional Kullback–Leibler (KL) divergence objective (Gong et al., 2023):

$$\mathcal{L}_{\text{struct}} = \lambda \text{KL}(P\|Q) + (1-\lambda)\text{KL}(Q\|P), \tag{11}$$

where $\text{KL}(P\|Q) = \sum_i \sum_{j\neq i} p_{j|i} \log \frac{p_{j|i}}{q_{j|i}}$. The hyperparameter $\lambda \in [0,1]$ (defaulted to 0.1) balances the trade-off between false positives and missing structural features. Detailed derivations are provided in Appendix C.1.

**Value-weighted MSE.** While $\mathcal{L}_{\text{struct}}$ preserves the global topology, precise distance estimation requires an additional supervised term. Given that Hi-C data exhibits higher reliability for smaller spatial distances (corresponding to high-intensity contacts), we introduce a value-weighted mean squared error (MSE) loss (Wang et al., 2024). This loss assigns adaptive weights based on the rank of true distance values rather than treating all errors uniformly.

For a batch containing distinct distance values, we compute weights $w_v$ for each value $v \in \mathcal{V}$ (see Appendix C.2 for details). The weighted MSE is defined as:

$$\mathcal{L}_{\text{weighted\_mse}} = \sum_{v \in \mathcal{V}} w_v \cdot \frac{1}{N_v} \sum_{(i,j) \in \mathcal{I}_v} (y_{ij} - \hat{y}_{ij})^2, \tag{12}$$

where $\mathcal{I}_v$ denotes the set of bin pairs with target distance $v$, $N_v = |\mathcal{I}_v|$, $y_{ij}$ is the target distance derived from $IF_{ij}$, and $\hat{y}_{ij} = \|\hat{\mathbf{s}}_i - \hat{\mathbf{s}}_j\|$ represents the predicted Euclidean distance between bins $i$ and $j$.

## 3.5 STABILITY OF INERTIAL FRAME ALIGNMENT

During our experiments, we observed that inputs from physics- or regularization-based reconstructions (e.g., 3DMAX, LorDG) consistently benefited from inertial-frame alignment, whereas contact-matrix eigendecomposition methods (e.g., Gram) showed little or no gain. To explain this contrast, recall that the input coordinates are defined as $\mathbf{C}^* = \{(x_i^*, y_i^*, z_i^*)\}_{i=1}^N \in \mathbb{R}^{N\times 3}$. We define its sample covariance

$$\Sigma_{\mathbf{C}^*} = \frac{1}{N}(\mathbf{C}^*)^\top \mathbf{C}^*, \tag{13}$$

which admits eigenvalues $\mu_1 \geq \mu_2 \geq \mu_3$ and orthonormal eigenvectors $u_1, u_2, u_3$ corresponding to the principal axes. The spectral gap

$$\delta(\mathbf{C}^*) = \mu_1 - \mu_2 \tag{14}$$

quantifies how well the first principal direction is separated from the remainder.

Given two coordinate sets $\mathbf{C}^{*(1)}$ and $\mathbf{C}^{*(2)}$, we measure the angular difference of their leading inertial axes by

$$\theta_{\text{PC1}}\big(\mathbf{C}^{*(1)}, \mathbf{C}^{*(2)}\big) = \arccos\big(\big|u_1(\mathbf{C}^{*(1)})^\top u_1(\mathbf{C}^{*(2)})\big|\big). \tag{15}$$

The stability of the principal directions under perturbations is controlled by the Davis–Kahan theorem (Davis & Kahan, 1970). Let $A$ be a symmetric matrix with leading eigenvector $u$ and spectral gap $\delta$. If $\widetilde{A} = A + \Delta A$ has leading eigenvector $\widetilde{u}$, then

$$\sin \angle\big(u, \widetilde{u}\big) \leq \frac{\|\Delta A\|_2}{\delta}, \tag{16}$$

where $\|\cdot\|_2$ denotes the spectral norm. Applied to $\Sigma_{\mathbf{C}^*}$ (Equation (13)), Equation (16) shows that when the spectral gap $\delta(\mathbf{C}^*)$ (Equation (14)) is small, even minor perturbations in the data can rotate the leading axis significantly. Conversely, a large gap yields stable inertial axes and reliable alignment. This analysis and explanation can be found in Appendix D.

## 4 EXPERIMENTS

**Datasets.** We evaluated our method on two single-cell Hi-C datasets: human frontal cortex cell (Wang & Cheng, 2024) and B-Lymphocyte cell (Oluwadare et al., 2020). Both datasets followed the

Table 1: Performance comparison of six methods on 3D chromosome structure reconstruction from single-cell Hi-C data (Frontal cortex cell test set). Metrics report distance-based Spearman correlation (dSCC ↑) and root mean square error (dRMSE ↓) at four resolutions. Best results in bold.

| Method | 320KB | | 160KB | | 80KB | | 40KB | |
|---|---|---|---|---|---|---|---|---|
| | dSCC↑ | dRMSE↓ | dSCC↑ | dRMSE↓ | dSCC↑ | dRMSE↓ | dSCC↑ | dRMSE↓ |
| 3DMAX | 0.2780 | 23.1587 | 0.2302 | 23.3439 | 0.1774 | 23.9174 | 0.1754 | 24.6538 |
| LorDG | 0.6681 | 92.6582 | 0.6997 | 102.1392 | 0.6342 | 100.0507 | 0.5841 | 96.1048 |
| HiC-GNN | 0.2432 | 0.8366 | 0.2077 | 0.9083 | 0.1370 | 0.9352 | 0.0915 | 0.9456 |
| HiCEGNN | 0.5804 | 0.2744 | 0.5351 | 0.3550 | 0.3288 | 0.4158 | 0.2506 | 0.4317 |
| IG-3DMAX | **0.9006** | **0.1697** | **0.8577** | **0.1835** | **0.7727** | **0.2192** | **0.7187** | **0.2410** |
| IG-LorDG | **0.8713** | **0.1544** | **0.8056** | **0.1997** | **0.6835** | **0.2398** | **0.6036** | **0.2574** |

chromosome partitioning scheme of Wang & Cheng (2025), with training sets (frontal cortex: chr 1,3,5,7,8,9,11,13,15,16,17,19,21,22; B-Lymphocyte: adds chr 23), validation sets (chr 2,6,10,12 for both), and test sets (chr 4,14,18,20 for both).

**Baselines and Implementation.** We compare our method with both classical numerical and deep learning baselines for 3D chromosome structure reconstruction. The classical numerical methods include 3DMax (Oluwadare et al., 2018) and LorDG (Trieu & Cheng, 2017), while the deep learning baselines include HiC-GNN (Hovenga et al., 2023) and HiCEGNN (Wang & Cheng, 2025). All baselines were run with their default configurations as provided in the respective source codes.

**Metrics.** We evaluate 3D chromosome reconstruction using two metrics: (1) **Distance Spearman correlation coefficient (dSCC)** (Oluwadare et al., 2018) measures rank correlation between predicted and ideal distances (range [-1,1]); higher values indicate better structural quality and scale invariance. (2) **Distance root mean square error (dRMSE)** (Varoquaux et al., 2014) quantifies absolute distance errors; lower values denote higher accuracy and similarity to ideal distance map.

### 4.1 MAIN RESULTS

**Reconstruction performance on Frontal cortex cell dataset.** Table 1 compares six methods on single-cell Hi-C data. Our two variants, IG-3DMAX and IG-LorDG, consistently outperform all baselines in both dSCC and dRMSE across all resolutions. For example, at 320 kB, IG-3DMAX achieves a dSCC of 0.9006, significantly higher than HiCEGNN (0.5804) and 3DMAX (0.2780), while reducing dRMSE to 0.1697 from 0.2744. Similar improvements are observed at other resolutions, with dSCC gains often exceeding 50% and dRMSE reductions of 30–40%. These results demonstrate that combining inertial-frame canonicalization with our Transformer yields state-of-the-art accuracy and stronger resolution-agnostic performance compared to equivariant and numerical baselines. Traditional numerical methods (3DMAX, LorDG) perform substantially worse, with dRMSE values orders of magnitude higher, due to the lack of direct di stance supervision. Gram-matrix inputs also yield poor performance in our pipeline; we analyze this instability in Appendix D.

**Reconstruction performance on B-Lymphocyte cell dataset.** Table 2 compares six methods on single-cell Hi-C data. IG-3DMAX achieves the best dSCC and lowest dRMSE across all four resolutions, demonstrating robust superiority. For example, at 1MB it attains a dSCC of 0.9209 and dRMSE of 0.0822, outperforming all baselines. Similar advantages are observed at finer resolutions (500KB, 250KB, 100KB), with dSCC consistently above 0.87 and dRMSE below 0.08. IG-LorDG shows competitive dSCC at 500KB (0.8367) and 250KB (0.8440), but weaker performance at other resolutions. This variation stems from LorDG's inherent reconstruction limitations on this dataset (*e.g.*, high dRMSE 107.7091 at 1MB), which constrain IG-LorDG's input quality. In contrast, IG-3DMAX's stability highlights our framework's resilience to input variations, delivering state-of-the-art accuracy regardless of baseline method performance.

### 4.2 ABLATION STUDIES

**Learning rate robustness.** We evaluate our method's robustness to input quality by varying the learning rates $\{1, 0.5, 0.1, 0.05, 0.01\}$ of 3DMAX and LorDG on Frontal cortex data. Input coordi-

Table 2: Performance comparison of six methods on 3D chromosome structure reconstruction from single-cell Hi-C data (B-Lymphocyte cell test set). Metrics report distance-based Spearman correlation (dSCC ↑) and root mean square error (dRMSE ↓) at four resolutions. Best results in bold.

| Method | 1MB | | 500KB | | 250KB | | 100KB | |
|--------|-----|-----|-------|-----|-------|-----|-------|-----|
| | dSCC↑ | dRMSE↓ | dSCC↑ | dRMSE↓ | dSCC↑ | dRMSE↓ | dSCC↑ | dRMSE↓ |
| 3DMAX | 0.9131 | 146.7464 | 0.8603 | 139.7606 | 0.8148 | 136.9766 | 0.6548 | 127.9119 |
| LorDG | 0.7462 | 107.7091 | 0.8103 | 117.5711 | 0.8316 | 112.0299 | 0.8395 | 98.9512 |
| HiC-GNN | 0.6778 | 0.2570 | 0.6457 | 0.2373 | 0.5827 | 0.1312 | 0.5334 | 0.2089 |
| HiCEGNN | 0.8847 | 0.0839 | 0.8068 | 0.0838 | 0.7530 | 0.0823 | 0.8017 | 0.0795 |
| IG-3DMAX | **0.9209** | **0.0822** | **0.9081** | **0.0777** | **0.8861** | **0.0593** | **0.8708** | **0.0790** |
| IG-LorDG | 0.8413 | 0.1114 | **0.8367** | 0.0979 | **0.8440** | **0.0675** | 0.7939 | 0.0867 |

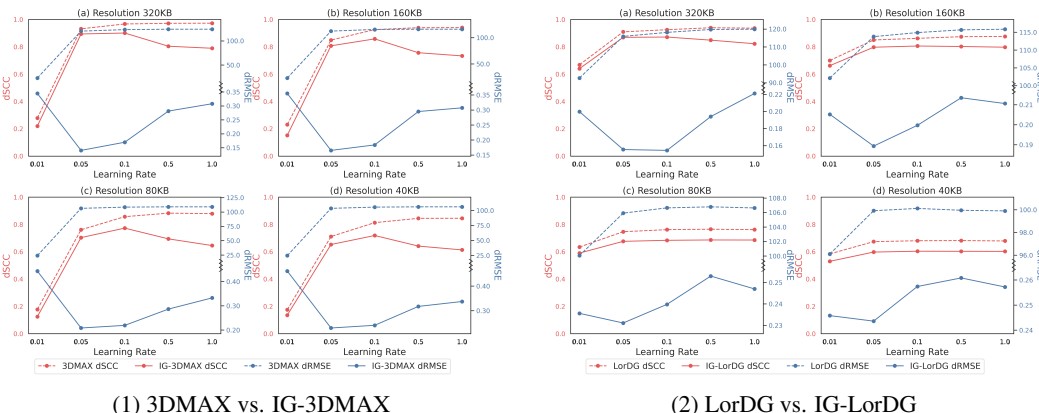

(1) 3DMAX vs. IG-3DMAX      (2) LorDG vs. IG-LorDG

Figure 3: Learning rate ablation: performance at four resolutions for five learning rates. Red: dSCC (left axis). Blue: dRMSE (right axis).

nates generated under each setting are evaluated at four resolutions using dSCC (↑) and dRMSE (↓). Figure 3 shows: (1) 3DMAX vs. IG-3DMAX, (2) LorDG vs. IG-LorDG. Our IG variants maintain consistently high dSCC while reducing dRMSE to $10^{-1}$ scale across all learning rates. Baselines show significant dRMSE fluctuations (up to $10^2$ scale) and unstable performance. This demonstrates InertialGenome's superior stability and accuracy regardless of input learning rate and resolution.

**Loss Components** We analyze the contribution of the structural stability loss ($\mathcal{L}_{\text{struct}}$) versus the coordinate regression loss ($\mathcal{L}_{\text{weighted\_mse}}$) by varying their weighting ratio $\alpha/\beta$ in the total objective (see Appendix E.1).

**Component ablation.** We assess the impact of key design choices by removing inertial-frame alignment, RoPE-3D, or Nyström encoding from our full model (see Appendix E.2).

## 4.3 CASE STUDIES

**TAD domain consistency validation.** We assess whether bin within the same TAD are spatially closer than those across TADs (see Appendix F.1). Figure 4 shows intra- vs. inter-TAD distances for IG-3DMAX and HiCEGNN. IG-3DMAX yields consistently shorter intra-TAD distances across chromosomes 4, 14, 18, and 20, with intra/inter ratios of 0.76–0.80 and highly significant $p$-values (Appendix Table 8). HiCEGNN shows higher ratios (0.91–0.99) and weaker significance (*e.g.*, $p = 0.159$ for chr20), indicating poor domain compaction. These results confirm that IG-3DMAX better captures TAD-level spatial organization than HiCEGNN.

**A/B compartment validation.** We validate the biological plausibility of our reconstructed structures using A/B compartment analysis (see Appendix F.2). Appendix Figure 6 compares distance distributions for IG-3DMAX and HICEGNN. IG-3DMAX shows significantly shorter intra-compartment (A–A, B–B) than inter-compartment (A–B) distances ($p_{\text{A}} = 0.0001$, $p_{\text{B}} = 0.0038$), confirming expected compartmental organization. In contrast, HICEGNN shows no significant

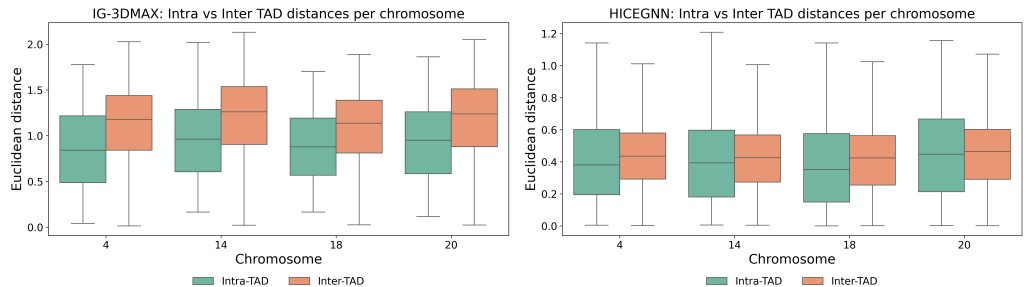

Figure 4: Intra- vs. inter-TAD Euclidean distances for IG-3DMAX (left) and HiCEGNN (right). Lower intra-TAD distances indicate stronger spatial clustering.

Table 3: Cross-resolution transfer results. Bold: improvement over same-resolution original model; underline: HICEGNN transfer improvement.

| Method | 320kb → 160kb | | 320kb → 80kb | | 320kb → 40kb | |
|---|---|---|---|---|---|---|
| | dSCC ↑ | dRMSE ↓ | dSCC ↑ | dRMSE ↓ | dSCC ↑ | dRMSE ↓ |
| HICEGNN-Transfer | 0.5815 | 0.2997 | 0.4080 | 0.4108 | 0.2343 | 0.3840 |
| HICEGNN-Original | 0.5351 | 0.3550 | 0.3288 | 0.4158 | 0.2506 | 0.4317 |
| IG-3DMAX-Full | **0.7455** | 0.3159 | **0.6766** | 0.3464 | **0.6528** | 0.3588 |
| IG-3DMAX-Selective | **0.7399** | 0.3089 | **0.6698** | 0.3447 | **0.6467** | 0.3567 |
| IG-3DMAX-Separate | **0.7434** | **0.3057** | **0.6714** | 0.3392 | **0.6480** | 0.3580 |
| IG-3DMAX-Original | 0.7332 | 0.3071 | 0.6451 | 0.3328 | 0.6132 | 0.3374 |

A–A/A–B separation ($p_A = 0.4360$) and weak B–B separation ($p_B = 0.0000$), indicating poor compartmentalization. These results demonstrate that IG-3DMAX better captures spatial compartment segregation than HICEGNN.

**Cross-resolution transfer learning.** We evaluate transfer from 320kb to finer resolutions (160kb, 80kb, 40kb). Table 3 shows IG-3DMAX consistently outperforms both its original model and HICEGNN variants in dSCC and dRMSE. At higher resolutions (80kb, 40kb), IG-3DMAX maintains stable dRMSE ($10^{-2}$) and improves dSCC by 5%, while HICEGNN degrades (dSCC drops at 40kb). Our gains stem from RoPE and inertial-frame alignment, which preserve spatial relations across scales. HICEGNN lacks such geometric adaptation, leading to unstable performance. IG-3DMAX demonstrates robust, resolution-agnostic reconstruction, with advantages magnified at finer resolutions.

# 5  CONCLUSION

We presented InertialGenome, a Transformer-based framework for robust, resolution-agnostic 3D chromosome reconstruction. It uses inertial-frame canonicalization for pose invariance and a geometry-aware Transformer with Nyström positional encoding to capture long-range interactions efficiently. Experiments on two single-cell datasets across four resolutions show that InertialGenome consistently outperforms classical (3DMAX, LorDG) and deep learning (HiC-GNN, HiCEGNN) methods in both structural accuracy and functional validation. Our cross-resolution strategy further boosts high-resolution performance by up to 5%, demonstrating strong generalization and biological plausibility.

By decoupling physical constraints from architecture, InertialGenome offers a flexible alternative to SE(3)-equivariant models, paving the way for scalable 3D genome modeling. Future work will integrate multi-modal genomic data to enhance reconstruction robustness across diverse experimental conditions.

ETHICS STATEMENT

Our study complies strictly with the ICLR Code of Ethics. The research involves no human partici-pants or sensitive personal data. All datasets and code employed adhere to their respective licensing agreements. This work presents foundational research that does not raise concerns related to fair-ness, privacy, security, or potential misuse. We affirm that all ethical aspects have been comprehen-sively considered and addressed.

REPRODUCIBILITY STATEMENT

We are dedicated to facilitating the reproducibility of our work. Full details required to repli-cate our key findings—including data access instructions, experimental parameters, model archi-tectures, evaluation metrics, and model checkpoints—are provided in our GitHub repository at `https://github.com/yize1203/InertialGenome`. Users can utilize our detailed doc-umentation and code scripts to faithfully reproduce the results, ensuring research transparency and methodological rigor.

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

## THE USE OF LARGE LANGUAGE MODELS

**Answer:** In the preparation of this work, the authors used large language models (LLMs) for two specific purposes: (1) to refine and polish English language expression, and (2) to assist in the formulation and typesetting of mathematical equations in LaTeX. All scientific content, research design, analysis, and conclusions remain solely the responsibility of the authors.

## A    RELATED WORK

### A.1    DISTANCE AND CONTACT CONSTRAINED OPTIMIZATION METHODS

Distance-based methods rely on the assumption that genomic loci closer in the one-dimensional sequence are also spatially proximal in three dimensions. The central idea is to exploit the inverse relationship between Hi-C contact frequency and physical distance, thereby transforming the contact matrix into a distance matrix and reformulating the reconstruction task as inferring 3D coordinates from pairwise distances. A variety of algorithms adopt this paradigm, including miniMDS (Rieber & Mahony, 2017), 3DMax (Oluwadare et al., 2018), ChromSDE (Zhang et al., 2013), Chromosome3D (Adhikari et al., 2016), ShNeigh1 (Li et al., 2020), LorDG (Trieu & Cheng, 2017), and others. These methods aim to faithfully recover the underlying geometric distances. For example, 3DMax (Oluwadare et al., 2018) assumes that Hi-C data follow a Gaussian distribution and that contact counts are independent given the structure, defining a log-likelihood objective to identify the most probable conformation. LorDG (Trieu & Cheng, 2017), in contrast, employs a nonlinear Lorentzian function to enforce structural consistency while improving robustness to noisy distance constraints.In contrast, contact-based methods (Meluzzi & Arya, 2013; Trieu & Cheng, 2014; Abbas et al., 2019) directly translate interaction frequencies (IFs) into spatial constraints without first converting them into distances. For example, GEM (Abbas et al., 2019) enforces distance thresholds between chromosome segments and incorporates biophysical feasibility to reconstruct 3D structures.

### A.2    PROBABILITY BASED METHODS

Probabilistic methods typically assume that the global 3D structure underlies the observed contact map and formulate chromosome reconstruction as a Bayesian inference or maximum likelihood problem. In this framework, the contact matrix is modeled as data generated from specific probability distributions, with representative approaches including BACH (Hu et al., 2013), MCMC5C (Rousseau et al., 2011), PASTIS (Varoquaux et al., 2014), PGS (Hua et al., 2018), and CHROM-STRUCT 4 (Caudai et al., 2018). For instance, MCMC5C (Rousseau et al., 2011) employs a Gaussian prior and uses Markov Chain Monte Carlo (MCMC) sampling to infer spatial coordinates from the posterior distribution of interaction frequencies. BACH (Hu et al., 2013) assumes a Poisson distribution for contact counts and applies MCMC to sample chromosome conformations. PASTIS (Varoquaux et al., 2014), in contrast, optimizes spatial coordinates by maximizing the likelihood under the assumption that contact frequencies follow a Poisson distribution conditioned on 3D positions.

### A.3    DEEP LEARNING BASED METHODS

Unlike traditional distance and probability-based approaches, deep learning offers a fundamentally different paradigm for reconstructing 3D chromosome structures from Hi-C data. HiC-GNN (Hovenga et al., 2023) pioneered this line of work by applying graph convolutional networks to capture neighborhood structures in Hi-C interaction graphs, directly inferring 3D conformations from large-scale data. Recently, HiCEGNN (Wang & Cheng, 2025) extended this idea by employing an SO(3)-equivariant graph neural network (EGNN) to account for translational and rotational symmetries, enabling accurate prediction of the 3D coordinates of genomic loci. These graph-based deep learning approaches demonstrate not only the feasibility but also the advantages of deep learning for 3D genome modeling: they are faster and more straightforward than traditional optimization methods, and they can leverage large Hi-C datasets to capture common structural patterns often missed by conventional approaches. Building on this line of work, we propose a new deep learning framework. In contrast to existing graph-based methods, our approach demonstrates that a Transformer architecture can provide superior modeling capacity for 3D chromosome structure prediction.

## B  DERIVATION OF 3D GEOMETRIC POSITION EMBEDDING

We aim to extend the rotary position embedding (RoPE) mechanism to 3D Euclidean space for chromosome structure modeling. The goal is to design a geometric position embedding such that the relative rotation between two positions satisfies:

$$R_{x_1,y_1,z_1}^\top R_{x_2,y_2,z_2} = R_{x_1-x_2,y_1-y_2,z_1-z_2}. \tag{17}$$

To achieve this, we first recall that any imaginary number $z = a + bi$ can be written in polar form as:

$$z = r\cos\theta + ir\sin\theta = re^{i\theta}, \tag{18}$$

where $r = |z|$ and $\theta = \arg(z)$. This can further be expressed in matrix exponential form using the identity:

$$r\exp(\theta J) \equiv r\exp(\theta) \cdot J, \tag{19}$$

with $J = \begin{bmatrix} 0 & -1 \\ 1 & 0 \end{bmatrix}$, which satisfies $J^2 = -I$, $J^3 = -J$, $J^4 = I$. Thus,

$$r\exp(\theta J) = r\sum_{n=0}^{\infty} \frac{(\theta J)^n}{n!} = r\left(I + \theta J - \frac{\theta^2}{2!}I - \frac{\theta^3}{3!}J + \cdots\right). \tag{20}$$

Grouping terms by even and odd powers:

$$r\exp(\theta J) = r\left(\sum_{n=0}^{\infty} \frac{(-1)^n \theta^{2n}}{(2n)!}I\right) + r\left(\sum_{n=0}^{\infty} \frac{(-1)^n \theta^{2n+1}}{(2n+1)!}J\right) \tag{21}$$

$$= r\cos\theta \cdot I + r\sin\theta \cdot J \tag{22}$$

$$= r\begin{bmatrix} \cos\theta & -\sin\theta \\ \sin\theta & \cos\theta \end{bmatrix}. \tag{23}$$

Hence, the 2D rotation matrix can be written as $R_m = \exp(mJ)$, and it follows that:

$$R_m^\top R_n = \exp(-mJ)\exp(nJ) = \exp((n-m)J) = R_{n-m}. \tag{24}$$

Now, we generalize this to 3D. We define a 3D rotation operator $R_{x,y,z}$ as the exponential of a skew-symmetric matrix:

$$R_{x,y,z} = \exp\left(\theta \begin{bmatrix} 0 & -x & 0 & 0 & 0 & 0 \\ x & 0 & 0 & 0 & 0 & 0 \\ 0 & 0 & 0 & -y & 0 & 0 \\ 0 & 0 & y & 0 & 0 & 0 \\ 0 & 0 & 0 & 0 & 0 & -z \\ 0 & 0 & 0 & 0 & z & 0 \end{bmatrix}\right), \tag{25}$$

which decomposes into three independent 2D rotations along the $(x,y)$, $(y,z)$, and $(z,x)$ planes.

Expanding this matrix exponential yields:

$$R_{x,y,z} = \begin{bmatrix} \cos x\theta & -\sin x\theta & 0 & 0 & 0 & 0 \\ \sin x\theta & \cos x\theta & 0 & 0 & 0 & 0 \\ 0 & 0 & \cos y\theta & -\sin y\theta & 0 & 0 \\ 0 & 0 & \sin y\theta & \cos y\theta & 0 & 0 \\ 0 & 0 & 0 & 0 & \cos z\theta & -\sin z\theta \\ 0 & 0 & 0 & 0 & \sin z\theta & \cos z\theta \end{bmatrix}. \tag{26}$$

When applied to a 6-dimensional token embedding $\mathbf{q} = [q_0, q_1, q_2, q_3, q_4, q_5]^\top$, we obtain:

$$R_{x,y,z}\mathbf{q} = \begin{bmatrix} q_0 \\ q_1 \\ q_2 \\ q_3 \\ q_4 \\ q_5 \end{bmatrix} \odot \begin{bmatrix} \cos x\theta_0 \\ \cos x\theta_0 \\ \cos y\theta_0 \\ \cos y\theta_0 \\ \cos z\theta_0 \\ \cos z\theta_0 \end{bmatrix} + \begin{bmatrix} -q_1 \\ q_0 \\ -q_3 \\ q_2 \\ -q_5 \\ q_4 \end{bmatrix} \odot \begin{bmatrix} \sin x\theta_0 \\ \sin x\theta_0 \\ \sin y\theta_0 \\ \sin y\theta_0 \\ \sin z\theta_0 \\ \sin z\theta_0 \end{bmatrix}, \tag{27}$$

where $\theta_0$ is the base frequency (e.g., $10000^{-2/6}$), and $\odot$ denotes element-wise multiplication.

This construction ensures that the inner product satisfies:

$$(R_{x_1,y_1,z_1}\mathbf{q})^\top (R_{x_2,y_2,z_2}\mathbf{k}) = \mathbf{q}^\top R_{x_1-x_2,y_1-y_2,z_1-z_2}\mathbf{k}, \tag{28}$$

as required. Therefore, our 6D geometric embedding naturally captures relative spatial relationships through rotational invariance.

## C    DETAILED LOSS FUNCTIONS

### C.1    STRUCTURAL-LEARNING LOSS

Let the spatial distance representation be

$$D = \{D_1, D_2, ..., D_n\}, \quad D_i = \{D_{i,1}, D_{i,2}, ..., D_{i,n}\}, \tag{29}$$

and the 3D coordinate representation be

$$S = \{S_1, S_2, ..., S_n\}, \quad S_i = (x_i, y_i, z_i). \tag{30}$$

**Neighborhood Probability Distributions.**  Based on heterogeneity between genomic bins, the probabilities are:

$$p_{j|i} = \frac{\exp(-D_{ij})}{\sum_{k \neq i} \exp(-D_{i,k})}, \quad q_{j|i} = \frac{\exp(-\|\mathbf{s}_i - \mathbf{s}_j\|^2)}{\sum_{k \neq i} \exp(-\|\mathbf{s}_i - \mathbf{s}_k\|^2)}, \tag{31}$$

where $\|\mathbf{s}_i - \mathbf{s}_j\|^2$ is the squared Euclidean distance between bins $i$ and $j$.

**Bidirectional KL Divergence.**  Let $P = \{P_1, ..., P_n\}$ be the distance distribution and $Q = \{Q_1, ..., Q_n\}$ the 3D-space distribution. We define

$$KL(P\|Q) = \sum_i \sum_{j \neq i} p_{j|i} \log \frac{p_{j|i}}{q_{j|i}}, \tag{32}$$

$$KL(Q\|P) = \sum_i \sum_{j \neq i} q_{j|i} \log \frac{q_{j|i}}{p_{j|i}}. \tag{33}$$

Balancing false positives and misses via parameter $\lambda$, the final structural-learning loss is:

$$\begin{aligned} \mathcal{L}_{\text{struct}} &= \lambda\, KL(P\|Q) + (1-\lambda)\, KL(Q\|P) \\ &= \lambda \sum_i \sum_{j \neq i} p_{j|i} \log \frac{p_{j|i}}{q_{j|i}} + (1-\lambda) \sum_i \sum_{j \neq i} q_{j|i} \log \frac{q_{j|i}}{p_{j|i}}. \end{aligned} \tag{34}$$

### C.2    VALUE-WEIGHTED MSE LOSS

To emphasize smaller distances (high-intensity contacts), we weight the MSE by ranks. For a batch with $n$ distinct true distances, the weight for each value is computed as:

$$w_i = \frac{\text{rank}(i)}{n(n+1)/2}, \tag{35}$$

where $\text{rank}(i)$ is the ascending rank order of value $i$ (smallest value has rank 1 and receives the highest weight).

The final weighted MSE loss is:

$$\mathcal{L}_{\text{weighted-mse}} = \sum_{i \in \text{values}} w_i \cdot \frac{\sum (y_i - p_i)^2}{N_i}, \tag{36}$$

where $N_i$ is the frequency of true value $i$ in the current batch.

By combining Equation (34) and Equation (36), our model simultaneously learns accurate local distances and a consistent global structure, enabling more accurate and robust 3D chromosome reconstruction.

## D  DERIVATIONS FOR INERTIAL ALIGNMENT STABILITY

We summarize here the derivation and the empirical checks that complement the main text.

**Notation.** $C \in \mathbb{R}^{N \times N}$ denotes the contact matrix, $D = f(C)$ its corresponding distance matrix obtained through the monotone transform $f(\cdot)$, $H = I - \frac{1}{N}\mathbf{1}\mathbf{1}^\top$ the centering matrix, $B = -\frac{1}{2}HD^2H$ the Gram matrix, and $X \in \mathbb{R}^{N \times 3}$ the centered 3D coordinates whose sample covariance is $\Sigma_X = \frac{1}{N}X^\top X$.

**First-order perturbation.** Let $\Delta C$ be a small perturbation of $C$. By Taylor expansion of $f$ we have to first order

$$\Delta D \approx f'(C) \circ \Delta C.$$

Consequently, the Gram perturbation is

$$\Delta B \approx -H\left(D \circ \Delta D\right)H.$$

Taking operator norms and using $\|H\|_2 = 1$ yields the bound

$$\|\Delta B\|_2 \lesssim \|D\|_\infty \|f'(C)\|_\infty \|\Delta C\|_F.$$

Thus the size of $\Delta B$ scales linearly with the contact perturbation norm and with the current distances.

**Effect on principal components.** Let $u_1$ be the leading eigenvector of $B$ and $\widetilde{u}_1$ that of $B + \Delta B$. By the Davis–Kahan theorem (Equation (16) in the main text) we have

$$\sin \angle(u_1, \widetilde{u}_1) \leq \frac{\|\Delta B\|_2}{\delta_B},$$

where $\delta_B$ is the spectral gap of $B$ (difference between its first and second eigenvalues). Hence, even very small perturbations of $C$ can produce large rotations of the leading direction whenever $\delta_B$ is small; conversely, a large gap yields robust orientation.

**Empirical validation (Chromosome 3 at 320 kb).** We found through numerical calculations that many of the chromosome 3D coordinates used as input in grams are coplanar or almost coplanar, which directly indicates the problem of using grams as input. Of course, we still validated the difference between using two types of methods as inputs through our approach. We have selected chromosome 3 for validation here because it is a gram as the input that there is no coplanar sample. Table 4 reports the quantitative stability metrics of the Gram-based embedding and the 3DMax reconstruction for the same chromosome. We observe that the Gram top eigenvalues are very close (spectral gap $\delta \approx 0$), making its orientation unstable. 3DMax, on the other hand, shows a much larger spectrum spread ($\delta \approx 18.6$) and correspondingly stable orientation.

Table 4: Stability metrics for Chromosome 3 (320 kb).

|  | Top-3 spectrum | Rotation stability |
|---|---|---|
| Gram | [0.00547, 0.00546, 0.00502], $\delta = 0.0000$ | 2.83 |
| 3DMax | [70.62, 52.04, 41.30], $\delta = 18.58$ | 4.49e–5 |

**Noise experiment.** Figure 5 shows the median angle of the first principal component (PC1) as a function of injected noise scale. The Gram embedding (blue) displays rapid growth of the PC1 angle once noise is added, whereas the 3DMax embedding (orange) remains essentially unaffected. This is consistent with the theoretical bound above: a vanishing spectral gap makes the orientation of the Gram embedding highly sensitive to perturbations. These observations jointly validate our perturbation analysis: even when distances and neighborhood structure are perfectly preserved, a near-degenerate spectrum leads to unstable principal directions, whereas a large spectral gap confers rotational stability.

**IG-Gram experiment.** Table 5 illustrates the performance of Gram reconstruction as input into InertialGenome.

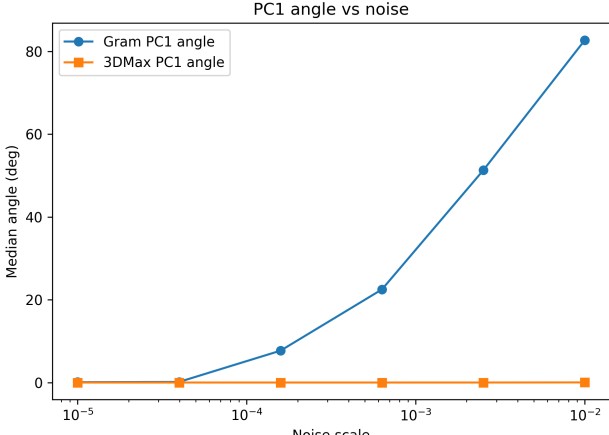

Figure 5: Median PC1 angle versus noise scale for the Gram and 3DMax.

Table 5: dSCC and dRMSE of IG-Gram on different resolutions.

| Method | 320KB | | 160KB | | 80KB | | 40KB | |
|---|---|---|---|---|---|---|---|---|
| | dSCC↑ | dRMSE↓ | dSCC↑ | dRMSE↓ | dSCC↑ | dRMSE↓ | dSCC↑ | dRMSE↓ |
| IG-Gram | 0.4770 | 0.3728 | 0.1079 | 0.4656 | 0.0524 | 0.4811 | 0.0083 | 0.5383 |

## E SUPPLEMENTARY FOR ABLATION STUDY

### E.1 LOSS COMPONENTS

We analyze the contribution of the structural stability loss ($\mathcal{L}_{\text{struct}}$) versus the coordinate regression loss ($\mathcal{L}_{\text{weighted\_mse}}$) by varying their weighting ratio $\alpha/\beta$ in the total objective $\mathcal{L}_{\text{total}} = \alpha\mathcal{L}_{\text{struct}} + \beta\mathcal{L}_{\text{weighted\_mse}}$, where $\beta = 1 - \alpha$. As shown in Table 6, removing structural supervision entirely ($\alpha = 0$) yields reasonable performance at coarse resolutions but leads to significant degradation in dRMSE at fine scales (e.g., 40kb), indicating poor geometric consistency. In contrast, using only structural loss ($\alpha = 1$) improves dRMSE at coarse resolutions but harms distance correlation (dSCC). A certain proportion of structural regularization ($\alpha = 0.1, 0.5$) consistently achieves the best trade-off across all resolutions, validating its role in enforcing biologically plausible 3D genome topology without sacrificing coordinate fidelity.

Table 6: Ablation study about IG-3DMAX on the loss weighting between structural stability and coordinate regression. Reported metrics: distance-based Spearman correlation (dSCC ↑) and root mean square error (dRMSE ↓) across four genomic resolutions. Best results per resolution are bolded.

| Ratio ($\alpha/\beta$) | 320KB | | 160KB | | 80KB | | 40KB | |
|---|---|---|---|---|---|---|---|---|
| | dSCC↑ | dRMSE↓ | dSCC↑ | dRMSE↓ | dSCC↑ | dRMSE↓ | dSCC↑ | dRMSE↓ |
| 0.0 / 1.0 | 0.9030 | 0.1728 | 0.8627 | 0.1935 | 0.7532 | 0.2152 | 0.7132 | 0.2410 |
| 0.1 / 0.9 | **0.9029** | **0.1696** | **0.8595** | **0.1848** | **0.7663** | **0.2197** | **0.7158** | **0.2407** |
| 0.5 / 0.5 | **0.9002** | **0.1671** | **0.8580** | **0.1879** | **0.7741** | **0.2266** | **0.7203** | **0.2445** |
| 1.0 / 0.0 | 0.8815 | 0.1453 | 0.8484 | 0.1775 | 0.7677 | 0.2297 | 0.7192 | 0.2788 |

### E.2 COMPONENT ABLATION

Results using IG-3DMAX in Table 7 show: (1) Without inertial alignment, dRMSE increases across all resolutions (e.g., from 0.1547 to 0.1641 at 320 kb), indicating its role in stabilizing global structure; (2) Removing RoPE consistently degrades both dSCC and dRMSE, confirming that relative positional encoding is essential for structural fidelity; (3) Disabling the Nyström branch leads to the

largest performance drop at fine scales (e.g., dRMSE rises by 0.0114 at 40 kb), demonstrating its critical contribution to modeling long-range pairwise distances. The full model achieves the best trade-off between structural consistency and coordinate accuracy, outperforming all ablated variants at every resolution.

Table 7: Ablation study of key components in our model on single-cell Hi-C data (Frontal cortex test set). Reported metrics: distance-based Spearman correlation (dSCC ↑) and root mean square error (dRMSE ↓) across four genomic resolutions. Best results per resolution are bolded.

| Model | 320KB | | 160KB | | 80KB | | 40KB | |
|---|---|---|---|---|---|---|---|---|
| | dSCC↑ | dRMSE↓ | dSCC↑ | dRMSE↓ | dSCC↑ | dRMSE↓ | dSCC↑ | dRMSE↓ |
| Full (Ours) | **0.9030** | **0.1547** | **0.8621** | **0.1809** | **0.7757** | **0.2035** | **0.7297** | **0.2382** |
| w/o Inertial | 0.9008 | 0.1641 | 0.8598 | 0.1869 | 0.7737 | 0.2185 | 0.7226 | 0.2385 |
| w/o RoPE | 0.8976 | 0.1613 | 0.8566 | 0.1894 | 0.7709 | 0.2229 | 0.7213 | 0.2454 |
| w/o Nyström | 0.9002 | 0.1659 | 0.8607 | 0.1998 | 0.7746 | 0.2218 | 0.7214 | 0.2496 |

# F    EXPERIMENTAL VALIDATION

## F.1    VALIDATION VIA TAD DOMAIN CONSISTENCY

Topologically Associating Domains (TADs) are contiguous genomic regions within which loci tend to interact more frequently with each other than with loci outside the domain (Dixon et al., 2012). Validating the biological plausibility of predicted 3D chromosome structures can therefore be performed by examining whether spatial distances among loci within the same TAD are significantly shorter than those between loci from different TADs.

**TAD Boundary Acquisition** TAD regions can be obtained from public annotations (e.g., ENCODE project or 3D Genome Browser) or identified directly from Hi-C contact matrices using established TAD callers such as: TopDom (Shin et al., 2016), Armatus (Filippova et al., 2014), and HiCExplorer (Wolff et al., 2020). In our workflow, TADs were identified using TopDom following the procedure in Serra et al. (2017).

**Mapping 3D coordinates to TADs:** Each bin in the 3D structure corresponds to a genomic interval based on Hi-C resolution (e.g., 250 kb). For a bin indexed by $i$, its genomic span is $[i \times 250\,\text{kb}, (i + 1) \times 250\,\text{kb})$. We assign each 3D point to a TAD by checking whether its genomic start coordinate falls within the start and end of any TAD interval.

**Evaluation Metric (Mean intra-/inter-TAD distance):** For each TAD, we compute the average pairwise Euclidean distance $D_{\text{intra}}$ among all loci inside the same TAD. We also compute the average distance $D_{\text{inter}}$ between loci across different TADs. A well-structured 3D prediction should satisfy:

$$\text{mean}(D_{\text{intra}}) \ll \text{mean}(D_{\text{inter}}), \tag{37}$$

indicating spatial clustering of loci within the same TAD, consistent with known biological organization.

**Statistical Testing** We assessed whether intra-TAD distances are significantly smaller than inter-TAD distances using the Mann–Whitney U test. Table 8 summarizes the mean intra- and inter-TAD distances, intra/inter ratios, and corresponding p-values for IG-3DMAX and IG-LorDG reconstructions across chromosomes 4, 14, 18, and 20.

These metrics together provide a quantitative assessment of whether the reconstructed 3D structures preserve known domain-level chromatin organization. In particular, while IG-3DMAX consistently shows significantly lower intra-TAD distances with low intra/inter ratios across all chromosomes, HiCEGNN exhibits weaker separation and a non-significant result for chromosome 20, indicating comparatively less biologically plausible domain organization in that case.

## F.2    VALIDATION VIA A/B COMPARTMENTALIZATION

To assess the biological plausibility of our predicted 3D chromosome structures, we validate them based on A/B compartment organization. In eukaryotic nuclei, chromosomes spatially segregate

Table 8: TAD-based validation statistics for IG-3DMAX and HiCEGNN reconstructions. Mean intra- and inter-TAD distances, intra/inter ratio, and Mann–Whitney U test $p$-values are reported for selected chromosomes.

| Model | Chromosome | Intra-TAD | Inter-TAD | Ratio (Intra/Inter) | U test $p$-value |
|---|---|---|---|---|---|
| IG-3DMAX | Chr4 | 0.852 | 1.122 | 0.760 | 0 |
| IG-3DMAX | Chr14 | 0.957 | 1.203 | 0.796 | 8.95e-93 |
| IG-3DMAX | Chr18 | 0.882 | 1.083 | 0.814 | 1.55e-66 |
| IG-3DMAX | Chr20 | 0.941 | 1.177 | 0.800 | 2.83e-47 |
| HiCEGNN | Chr4 | 0.408 | 0.442 | 0.925 | 2.61e-34 |
| HiCEGNN | Chr14 | 0.405 | 0.425 | 0.952 | 1.09e-4 |
| HiCEGNN | Chr18 | 0.380 | 0.416 | 0.914 | 3.16e-10 |
| HiCEGNN | Chr20 | 0.451 | 0.454 | 0.993 | 0.159 |

into two major compartments—A (active) and B (inactive)—with loci from the same compartment (A–A or B–B) tending to be spatially closer than loci from different compartments (A–B).

We follow the process of Lieberman-Aiden et al. (2009) to assign compartment labels. For each chromosome, we compute the Pearson correlation matrix of the normalized Hi–C contact matrix, then extract the first principal component (PC1). Loci with positive PC1 values are assigned to compartment A, and those with negative PC1 values to compartment B.

**Distance computation and grouping.** From the predicted 3D coordinates we compute pairwise Euclidean distances and categorize them into three groups:

- *intra-A*: distances between loci within compartment A;
- *intra-B*: distances between loci within compartment B;
- *inter-AB*: distances between loci from compartments A and B.

**Statistical analysis and visualization.** We compare the distributions of these three groups using box plots and permutation test. A valid 3D structure should exhibit significantly shorter distances for intra-compartment pairs (A–A and B–B) than for inter-compartment pairs (A–B), indicating that the predicted structures preserve the compartmental spatial organization.

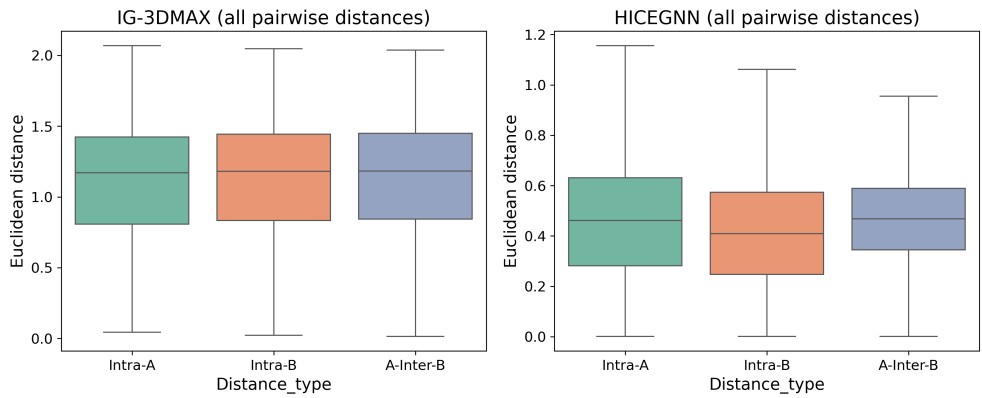

Figure 6: A/B compartment validation. Box plots show intra- and inter-compartment distances for IG-3DMAX (left) and HICEGNN (right).

### F.3 FISH-BASED VALIDATION OF 3D STRUCTURES

FISH experiments provide direct measurements of the spatial distances between genomic loci in the nucleus. To validate our 3D chromatin structure predictions, we compared the predicted distances between key regions—L1, L2, and L3—with the experimentally observed FISH distances reported by Rao et al.(Rao et al., 2014). These regions were identified as loop anchors using HiCCUPS

analysis, with L1 and L2 forming a strong interaction peak, while L3 served as a non-interactive control.

We evaluated our model outputs at $250\,\mathrm{kb}$ resolution, where sufficient structural detail is preserved to accurately localize these regions. For each chromosome, we computed the Euclidean distances between L1–L2 and L2–L3 in the predicted 3D structures, along with their corresponding contact probabilities from the original Hi-C data.

As shown in Table 9, the predicted L1–L2 distances are consistently shorter than the L2–L3 distances across all chromosomes, consistent with the experimental observations. Moreover, the contact probabilities exhibit an inverse trend: higher values for L1–L2 compared to L2–L3, reflecting stronger physical proximity. This agreement between predicted distances and Hi-C contact frequencies supports the biological plausibility of our modeled structures.

Table 9: FISH validation results on GM12878 chromosomes 11, 14, and 17 at $250\,\mathrm{kb}$ resolution. The table shows the predicted L1–L2 and L2–L3 distances, and the corresponding KR-normalized contact probabilities from Hi-C data.

| Chromosome | L1–L2 Distance | L1–L2 Probability | L2–L3 Distance | L2–L3 Probability |
|---|---|---|---|---|
| 11 | 0.8 | $2.74 \times 10^4$ | 3.3 | $3.92 \times 10^3$ |
| 14 | 2.2 | $9.10 \times 10^3$ | 13.1 | $2.07 \times 10^3$ |
| 17 | 2.3 | $2.05 \times 10^4$ | 3.7 | $1.32 \times 10^4$ |

The results demonstrate that our model captures the expected spatial organization: looped regions (L1–L2) are closer in space and exhibit higher contact frequencies than non-looped regions (L2–L3). This consistency with both FISH measurements and Hi-C data confirms that our method produces biologically realistic chromatin structures.

