# OpenReview forum: "A Resolution-Agnostic Geometric Transformer for Chromosome Modeling Using Inertial Frame"
_ICLR.cc/2026/Conference — ICLR 2026 Poster_

### Official Review · Reviewer_M6rw · 2025-10-22

**Soundness:** 3
**Presentation:** 3
**Contribution:** 2
**Rating:** 6
**Confidence:** 2

**Summary:**

This paper introduces InertialGenome, a novel Transformer-based framework for reconstructing 3D chromosome structures from Hi-C data. The core contributions are a two-stage process: first, it canonicalizes the initial 3D coordinates using an inertial frame to achieve pose-invariance, removing translational and rotational ambiguity. Second, it employs a geometry-aware Transformer, enhanced with 3D rotary position embeddings (RoPE) and Nyström-based structural features, to refine these coordinates. The proposed method demonstrates state-of-the-art performance on two single-cell datasets across four resolutions, showing particular strength in generalization and cross-resolution transfer learning.

**Strengths:**

The primary contribution is the elegant and effective use of inertial frame canonicalization as a preprocessing step for 3D chromosome structure refinement. This method directly addresses the fundamental problem of pose variance in geometric deep learning, providing a principled way to standardize inputs. This standardization appears to be the key driver behind the model's impressive resolution-agnostic capabilities and strong generalization performance.

The paper's novelty lies not in a single component, but in the intelligent synthesis of several ideas. While inertial frames and Transformers are not new in themselves, their application and combination in this domain are novel and well-motivated. The theoretical analysis of the inertial frame's stability (Section 3.5) using the Davis-Kahan theorem is a significant strength.

**Weaknesses:**

The most critical weakness is the omission of highly relevant and recent related work in both the discussion and the experimental comparison. This significantly impacts the claims of novelty and the comprehensiveness of the evaluation.

1. Missing Baseline: A key missing baseline is CHROMFORMER (NIPS'2022), which also uses a Transformer-based architecture for this exact problem. Without a direct comparison, it is difficult to ascertain whether the performance gains of InertialGenome stem from the novel inertial frame canonicalization or simply from the power of the Transformer backbone, which CHROMFORMER already established.

2. Unaddressed Novelty: The paper's geometry-aware positional encoding (3D-RoPE) is presented as a novel adaptation. However, the recent work "Learning the RoPEs: Better 2D and 3D Position Encodings with STRING (ICML'2025)" appears to propose a very similar 3D extension of RoPE. The lack of citation and discussion of this work obscures the precise novelty of the authors' formulation.

**Questions:**

1. Comparison with CHROMFORMER: Could the authors please comment on the CHROMFORMER (NIPS'2022) model? Given its use of a Transformer for the same task, it appears to be a crucial baseline. Can you provide a comparison, or a compelling argument for its exclusion, to better situate your performance results?

2. Novelty of 3D-RoPE: The recently proposed STRING (ICML'2025) introduced a similar 3D rotary position encoding. Could you please clarify the novelty of your 3D-RoPE implementation in relation to this prior work?

3. Ablation Study for Loss Parameter α: Your hybrid loss function in Equation (12) is controlled by the parameter α. Could you please provide an ablation study showing how performance (e.g., dSCC and dRMSE) varies with different values of α? This is important for understanding the interplay between the two loss components and for reproducibility.

4. In Section 3.4, you motivate the Value-Weighted MSE by stating that Hi-C data has higher reliability for smaller distances. However, another common approach for handling data with varying reliability or potential outliers is to use the Mean Absolute Error (MAE), which is inherently less sensitive to large errors than MSE. Could you please elaborate on the reasoning for choosing a weighted MSE scheme over a simpler, more robust loss function like MAE?

---

> ### Author Response · Authors · 2025-11-23
> **Response**
>
> **Q1:
> (1) Missing Baseline: A key missing baseline is CHROMFORMER.
> (2) Comparison with CHROMFORMER. Can you provide a comparison?**
>
> We thank the reviewer for this point. We implemented ChromFormer on our single-cell Hi-C benchmark, but its reliance on synthetic (Hi-C, 3D) pairs makes training increasingly slow at higher resolutions. We thus report results at 320 kb—the highest feasible resolution under comparable computational budgets.
>
> Crucially, our model significantly outperforms ChromFormer:
>
> | Model|320kb(dSCC$\uparrow$/dRMSE$\downarrow$)|
> | -------- | -------- |
> | Full (Ours)  | **0.9030/0.1547** |
> | ChromFormer | 0.5478/0.4179 |
>
> This confirms that our gains stem not from the Transformer backbone.
>
> For clarity, key distinctions between the two settings are summarized below:
>
> | Aspect | Bulk Hi-C Methods (e.g., ChromFormer, NeurIPS 2022) | Single-Cell Methods (e.g., InertialGenome, HiCEGNN) |
> |--------|---------------------------------------------------|----------------------------------------------------|
> | Input Data | Population-averaged (dense) bulk Hi-C | Extremely sparse single-cell Hi-C (<1% nonzeros) |
> | Ground Truth | Synthetic or limited FISH (e.g., yeast) | Hi-C-derived distances; full-chromosome dSCC/dRMSE |
> | Core Idea | Transformer trained on synthetic Hi-C → 3D pairs | Inertial-frame canonicalization + geometry-aware encoding |
> | Evaluation | Simulated structures or few FISH points | Real human scHi-C; held-out chromosomes; dSCC↑ / dRMSE↓ |
> | Technical Context | Standalone synthetic-data approach | Built on scHi-C reconstruction paradigm |
>
> **Q2:
> (1) Unaddressed Novelty: the recent work "STRING" is a very similar 3D extension of RoPE.
> (2) Novelty of 3D-RoPE:  How is your 3D-RoPE novel compared to STRING?**
>
> We thank the reviewer for noting STRING (ICML 2025). Our 3D-RoPE is analytic, parameter-free, and enforces distance-aware attention via fixed cosine–sine modulation—unlike STRING’s learnable, general-purpose design. It is also tightly integrated with inertial-frame canonicalization to embed chromatin-specific geometric priors.
>
> For clarity, we summarize the key distinctions in the table below:
>
> | Aspect | STRING | Our 3D-RoPE |
> |--------|--------|-------------|
> | Form | $R(\mathbf{r}) = \exp\left( \sum r_k L_k \right)$ (learnable $L_k$) | $R(\mathbf{r}) = \bigoplus_{k \in \{x,y,z\}} \mathrm{Rot}(r_k \theta_0)$ (fixed $\theta_0$) |
> | Parameters | Learnable generators ($L_k$) | No learnable parameters |
> | Philosophy | Data-driven, task-agnostic | Prior-driven, chromatin-specific |
> | Interpretability | Black-box (matrix exponential) | Explicit (inner products = relative displacements) |
>
>
> **Q3: Ablation Study for Loss Parameter α.**
>
> The overall training objective is a hybrid loss that combines a structural-preserving term and a value-weighted regression term:
>
> $L_{total} = \alpha L_{struct} + \beta L_{weighted\_mse}, \quad \beta = 1 - \alpha.$
>
> We performed sensitivity analysis by varying $\alpha/\beta$ in {0, 0.1, 0.5, 1.0}.
>
> | Ratio $\alpha/\beta$|320kb(dSCC$\uparrow$/dRMSE$\downarrow$)| 160kb(dSCC$\uparrow$/dRMSE$\downarrow$) | 80kb(dSCC$\uparrow$/dRMSE$\downarrow$) |40kb(dSCC$\uparrow$/dRMSE$\downarrow$)|
> | -------- | -------- | -------- |-------- |-------- |
> | 0.0 / 1.0  | 0.9030/0.1728 | 0.8627/0.1935 |0.7532/0.2152|0.7132/0.2410
> | 0.1 / 0.9  | **0.9029/0.1696** | **0.8595/0.1848** |**0.7663/0.2197**|**0.7158/0.2407**
> | 0.5 / 0.5  | **0.9002/0.1671** | **0.8580/0.1879** |**0.7741/0.2266**|**0.7203/0.2445**
> | 1.0 / 0.0  | 0.8815/0.1453 | 0.8484/0.1775 |0.7677/0.2297|0.7192/0.2788
>
> As shown in Table, removing structural supervision entirely ($\alpha = 0$) yields reasonable performance at coarse resolutions but leads to significant degradation in dRMSE at fine scales (e.g., 40kb), indicating poor geometric consistency. In contrast, using only structural loss ($\alpha = 1$) improves dRMSE at coarse resolutions but harms distance correlation (dSCC). A certain proportion of structural regularization ($\alpha = 0.1,0.5$) consistently achieves the best trade-off across all resolutions.
>
> Please see the updated manuscript (line 417-419 see Appendix G.1).
>
> **Q4: Why use weighted MSE instead of a more robust loss like MAE?**
>
> We thank the reviewer for this question. Weighted MSE better preserves biologically critical short-range distances by penalizing large errors more strongly.
>
> As shown below (320 kb, same weighting), weighted MSE consistently outperforms MAE when structural guidance is used ($\alpha \geq 0.1$):
>
> | Ratio $\alpha/\beta$|Weighted MSE(dSCC$\uparrow$/dRMSE$\downarrow$)| Weighted MAE(dSCC$\uparrow$/dRMSE$\downarrow$) |
> | -------- | -------- | --------
> | 0.0 / 1.0  | 0.9030/0.1728 | 0.9057/0.1854
> | 0.1 / 0.9  | 0.9029/0.1696 | 0.8953/0.1699
> | 0.5 / 0.5  |0.9002/0.1671 | 0.8996/0.1738
>
> Only when $\alpha = 0$ does MAE show a slight dSCC gain—but with worse distance accuracy (higher dRMSE). Thus, weighted MSE offers the best trade-off and is used in all results.

---

### Official Review · Reviewer_FLe5 · 2025-10-30

**Soundness:** 2
**Presentation:** 2
**Contribution:** 2
**Rating:** 2
**Confidence:** 2

**Summary:**

This model tackles the problem of inferring genomic 3D coordinates from a distance matrix derived from Hi-C data.  The approach involves training a transformer model to infer coordinates that, essentially, maximize the agreement between the observed distance matrix and the inferred one.  The novel aspects include using something called the Nystrom method to infer candidate sets of points from an RBF kernel matrix, and using RoPE embeddings in 3D.

**Strengths:**

The overall approach contains several novel ideas, as outlined above.

The paper contains fairly extensive empirical results, largely following the evaluation protocols proposed in previous work in this area.

**Weaknesses:**

Overall, I found the empirical validation of the methods pretty unsatisfying.  In general, it's difficult to benchmark 3D genome reconstruction methods, because it is hard to find an orthogonal source of gold standard structures.   One common approach is to aggregate the single-cell reconstructions and compare the result to a bulk Hi-C dataset from the same type of cells.  Another approach is it to take similar approach using imaging data (e.g., chromatin tracing).

The list of methods to infer 3D chromatin contacts (line 44) is radically incomplete.  You should focus on methods that infer genome-wide contacts, which would leave off 3C, 4C, and 5C, but would include GAM, SPRITE, ChIA-PET, and many others.

Similarly, the list of methods for 3D reconstruction (lines 72-77) is incomplete.  The list later on the same page is better.  You should re-organize the text so that all the related work is discussed at once.

The critique of related work is pretty vague.  You just say that all of these methods "rely on simplistic modeling of contact matrices as the sole input, lacking deeper structural interpretation, and their model expression ability may be constrained and limited."  It's not clear what you mean by "lacking deeper structural interpretation" nor what the "model expression" refers to.  I'd rather see specific critiques of particular methods, or at the very least a more precise critique.

line 110: Not every method uses these two steps.  You should make clear that the problem does not have to be solved in this fashion.  And even if it is broken down into two steps, there is not agreement in the field about the proper transfer function to translate from counts to distances.

Minor points:

The abstract should make it clear earlier (around line 15)  you are talking about single-cell Hi-C data.  You should also be careful to describe it as such in the intro to Section 4.  If you just say "Hi-C," that typically means bulk Hi-C.

line 107: The number of bins in the genome depends on what reference genome you use, not what cell line you are looking at.

line 116: What does "canonicalized" mean here?

line 216: Give the Williams & Seeger citation when you first mention the Nystrom method.

line 218: I thought the phrase "and the effectiveness of this approach has been confirmed by research" was oddly vague.  What kind of research?

A brief description of the Nystrom method (as in lines 218-221, but shorter) should appear in the Introduction.

**Questions:**

Why did you only use two single-cell Hi-C datasets?  There are many more available.

How did you decide which methods to compare your method against?  Did you run them yourself, or just compare to published results?

I don't actually understand the sentences (lines 81-83) that describe how an "inertial frame" is used.  Some definitions of terms would be helpful.  What do you mean by "inertial frame" here, and "position canonicalization"?

---

> ### Author Response · Authors · 2025-11-23
> **Response**
>
> **Q1: Overall, I found the empirical validation of the methods pretty unsatisfying.**
>
> We thank the reviewer for this thoughtful comment. We are also seeking better verification methods. In response, we added FISH-based validation on GM12878 cells using spatial distances from Rao et al. [1]. As shown in Appendix H.3, our model correctly predicts that loop anchors (L1–L2) are closer than control pairs (L2–L3), consistent with FISH data and the inverse Hi-C contact trend.
>
> We invite the reviewer to see Section H.3 for details.
>
> [1] Rao S S P, Huntley M H, Durand N C, et al. A 3D map of the human genome at kilobase resolution reveals principles of chromatin looping[J]. Cell, 2014, 159(7): 1665-1680.
>
> **Q2: You should include GAM, SPRITE, ChIA-PET, and many others.**
>
> We thank the reviewer for this suggestion. The Introduction now clearly distinguishes targeted (e.g., 3C, 4C, 5C) from genome-wide (e.g., Hi-C, GAM, SPRITE, ChIA-PET) chromatin conformation methods (lines 44–46).
>
> **Q3: Similarly, the list of methods for 3D reconstruction is incomplete.**
>
> We appreciate the reviewer’s observation. A concise overview of 3D reconstruction methods appears in the main text, while a detailed survey—including ChromSDE, miniMDS, ShRec3D, Pastis, BACH—is provided in Appendix B.
>
> **Q4: The critique of related work is pretty vague.**
>
> We have sharpened the critique to highlight that prior methods lack explicit geometric priors and that E(3)-equivariant models like HiCEGNN enforce excessive rotational symmetry, hindering representation of asymmetric structures such as CTCF loops. (see lines 80–83)
>
> **Q5: line 110: Not every method uses these two steps.**
>
> We appreciate the reviewer's comment. The power-law conversion $D_{ij} = IF_{ij}^{-\gamma}$ follows standard practice [2,3] and matches the preprocessing used by key baselines (e.g., HiC-GNN, HiCEGNN) to ensure fair comparison. Our contribution lies in the reconstruction architecture, not the distance mapping.
>
> [2] Pombo A, Nicodemi M. Physical mechanisms behind the large scale features of chromatin organization[J]. Transcription, 2014, 5(2): e28447.
>
> [3] Barbieri M, Chotalia M, Fraser J, et al. Complexity of chromatin folding is captured by the strings and binders switch model[J]. Proceedings of the National Academy of Sciences, 2012, 109(40): 16173-16178.
>
> **Q6: The abstract should make it clear earlier you are talking about single-cell Hi-C data.**
>
> We thank the reviewer for this crucial clarification. We have revised the abstract and Section 4 to explicitly use “single-cell Hi-C” at first mention, avoiding ambiguity with bulk Hi-C.
>
> **Q7: line 107: The number of bins in the genome depends on what reference genome you use**
>
> We thank the reviewer for this correction; lines 112–115 now clarify that the bin count (~248,947 at 1 kb) reflects the masked human reference genome, not a GM12878-specific value.
>
> **Q8: (1) line 116: What does "canonicalized" mean here? (2) I don't actually understand the sentences (lines 81-83) that describe how an "inertial frame" is used.**
>
> We thank the reviewer for pointing out the need for clearer definitions.
>
> We now define “canonicalized” as pose-normalization via alignment to the chromosome’s principal axes (line 122) and clarify that the inertial frame is derived from the 3D point cloud’s inertia tensor—a standard shape-canonicalization technique (lines 86–89).
>
> Please see the updated manuscript.
>
> **Q9: Give the Williams & Seeger citation when you first mention the Nystrom method.**
>
> I have cited this place, please review the manuscript again.
>
> **Q10: line 218: I thought the phrase "and the effectiveness of this approach has been confirmed by research" was oddly vague. What kind of research?**
>
> We agree with the reviewer that the original phrasing was vague. In the revised manuscript, we have removed the ambiguous statement and instead clearly describe how to use the Nyström method.
>
> **Q11: A brief description of the Nystrom method should appear in the Introduction.**
>
> We thank the reviewer for this suggestion. As requested, we have added a brief but precise description of the Nyström method in the Introduction (lines 92–94 of the revised manuscript).
>
> **Q12: Why did you only use two single-cell Hi-C datasets?**
>
> We appreciate the reviewer’s comment. We use the same two single-cell Hi-C datasets as HiC-GNN and HiCEGNN to ensure fair comparison, as these are the standard benchmarks in prior work. We agree broader validation is valuable and plan to include more datasets in future studies.
>
> **Q13: How did you decide which methods to compare your method against?**
>
> We thank the reviewer for this important question.
>
> We selected four representative baselines—3DMax, LorDG, HiC-GNN, and HiCEGNN—as they span classical and state-of-the-art deep learning approaches and are standard in prior work. All methods were re-run by us using their official code, default hyperparameters, and our shared data splits/preprocessing to ensure fair comparison.

---

### Official Review · Reviewer_YNMs · 2025-10-30

**Soundness:** 2
**Presentation:** 1
**Contribution:** 2
**Rating:** 4
**Confidence:** 4

**Summary:**

This paper introduces InertialGenome, a novel Transformer-based framework for 3D chromosome reconstruction that is designed to be robust to different resolutions. The author proposes an inertial frame canonicalization and geometry-aware positional encoding scheme that combines 3D RoPE and Nyström approximation. The method demonstrates strong empirical results, outperforming baselines.

**Strengths:**

1. The use of inertial frame canonicalization to achieve pose invariance is a reasonable and well-justified preprocessing step.

2. The model consistently outperforms existing baselines in reconstruction metrics, TAD consistency and cross-resolution generalization.

**Weaknesses:**

1. In Sec 3.2, the textual description (L180-183) appears to contradict the mathematical formulation in Eq 5, where 'Selective' and 'Separate' modes seem functionally identical.

2. In Eq 2, the query/key vectors are intended to be 6D vector. Can we understand six as a pair, just as we considered two as a pair in RoPE? This lack of clear explanation hinders understanding.

3. It is not explicitly stated, but is the 'vocab_size' simply the total number of bins for a given chromosome (248,947 at 1Kb resolution)? This confirmation would be helpful for understanding.

4. The paper claims RoPE-3D has limitations in 'capturing pairwise distance relationships' (L214), but this claim is not sufficiently substantiated or demonstrated. The paper should provide an ablation study to demonstrate the contribution of this Nyström approximation .

5. In Sec 3.2, $u_i$, $A$, and $O$ are used in the main text but are only clearly defined several lines later, which hinder understanding.

6. The paper relies exclusively on distance-based metrics (dSCC, dRMSE). For a task focused on 3D structure reconstruction, TM-score or RMSD are essential for evaluating the global structural similarity of the predicted conformations.

7. The paper claims the method is 'resolution-agnostic' and attributes this to the inertial-frame alignment and RoPE. This explanation is high-level and lacks a deep analysis. A more thorough explanation is needed to precisely understand why this specific combination of components grants robustness to resolution changes, which is a key claim of the paper.

**Questions:**

1. On Line 219, the phrase "relationships between anchor points" is used twice in the same sentence. Is this a typographical error?

---

> ### Author Response · Authors · 2025-11-22
> **Response**
>
> **Q1: In Sec.3.2, the textual description (L180–183) appears to contradict the mathematical formulation in Eq.(5).**
>
> We thank the reviewer for this observation. The two modes differ in how input halves are processed: Selective applies RoPE only to the spatial half and passes the feature half unchanged (parameter-free), while Separate uses independent projections for both halves. We have clarified this distinction in Section 3.2 (lines 194–215) and updated Equation 5 accordingly—please refer to the revised manuscript for details.
>
> **Q2:This lack of clear explanation hinders understanding In Eq 2.**
>
> We thank the reviewer for the comment. Our 3D RoPE extends standard 2D RoPE by assigning each of the three orthogonal rotation planes---$(x,y)$, $(y,z)$, and $(z,x)$---to an independent 2D rotary subspace, requiring six dimensions in total. This follows from the matrix exponential form of 3D rotations (see Appendix C). We have added a brief clarification in Section 3.2 (lines 171--192) and refer readers to the revised manuscript and appendix for details.
>
> **Q3: It is not explicitly stated, but is the 'vocab_size' simply the total number of bins for a given chromosome (248,947 at 1Kb resolution)? This confirmation would be helpful for understanding.**
>
> Yes. In our setting, 'vocab_size' corresponds to the total number of genomic bins (i.e., sequence tokens) for a chromosome at a given resolution (e.g., 248,947 bins at 1Kb resolution). Each bin is treated as a unique “token” with an integer ID in [0, vocab_size - 1].
>
> **Q4: Weakness 4 (the contribution of this Nyström approximation) and Weakness 7 (The paper claims the method is 'resolution-agnostic' and attributes this to the inertial-frame alignment and RoPE) mentioned issues related to ablation experiments, and we hereby provide a unified response.**
>
> **Q4.1. Contribution of  inertial frame canonicalization**
>
> **Q4.2. Contribution of ROPE**
>
> **Q4.3. Contribution of Nystrom Approximation**
>
> We conducte ablation experiments to address the Q4.1,4.2,4.3 mentioned above.
>
> | Model|320kb(dSCC$\uparrow$/dRMSE$\downarrow$)| 160kb(dSCC$\uparrow$/dRMSE$\downarrow$) | 80kb(dSCC$\uparrow$/dRMSE$\downarrow$) |40kb(dSCC$\uparrow$/dRMSE$\downarrow$)|
> | -------- | -------- | -------- |-------- |-------- |
> | Full (Ours)  | **0.9030/0.1547** | **0.8621/0.1809** |**0.7757/0.2035**|**0.7297/0.2382**
> | w/o Inertial | 0.9008/0.1641 | 0.8598/0.1869 |0.7737/0.2185|0.7226/0.2385
> | w/o RoPE     | 0.8976/0.1613 | 0.8566/0.1894 |0.7709/0.2229|0.7213/0.2454
> | w/o Nyström  | 0.9002/0.1659 | 0.8607/0.1998 |0.7746/0.2218|0.7214/0.2496
>
> We assess key components by ablating inertial alignment, RoPE-3D, or Nystr"om encoding in IG-3DMAX. Results show: (1) removing inertial alignment increases dRMSE across resolutions (e.g., 0.1547 → 0.1641 at 320 kb), confirming its role in global stability; (2) ablating RoPE degrades both dSCC and dRMSE, highlighting its importance for structural fidelity; and (3) disabling Nystr"om causes the largest drop at fine scales (e.g., +0.0114 dRMSE at 40 kb), underscoring its value in modeling long-range geometry. The full model consistently outperforms all variants.
>
> Please see the updated manuscript (line 420-421 see Appendix G.2).
>
> **Q5: In Sec 3.2, $u_i$, $A$, and $O$ are used in the main text but are only clearly defined several lines later, which hinder understanding.**
>
> Thank you for the suggestion. We have revised Section 3.2 (line 236-238) to define all symbols (e.g., $u_i$, $A_g$, and $O_g$) before their first use and improved the exposition for better clarity.
>
> **Q6: The paper relies exclusively on distance-based metrics (dSCC, dRMSE). should evaluate TM-score or RMSD.**
>
> We appreciate the suggestion. However, at single-cell resolution, true 3D coordinates are not experimentally available—structures are inferred from Hi-C data itself. Thus, the field widely uses distance-based metrics like dSCC and dRMSE to assess consistency between predicted distances and observed contacts [[1],[2]]. Metrics such as RMSD or TM-score require a reliable ground-truth structure, which does not exist for genome-scale chromatin (unlike proteins). Using any single reconstructed conformation as “ground truth” would be biased, as many distinct 3D structures can explain the same Hi-C map . Hence, dSCC/dRMSE remain the standard and most meaningful evaluation protocols.
>
> [1] Oluwadare O, Zhang Y, Cheng J. A maximum likelihood algorithm for reconstructing 3D structures of human chromosomes from chromosomal contact data[J]. BMC genomics, 2018, 19(1): 161.
>
> [2] Hovenga V, Kalita J, Oluwadare O. Hic-gnn: A generalizable model for 3d chromosome reconstruction using graph convolutional neural networks[J]. Computational and Structural Biotechnology Journal, 2023, 21: 812-836.
>
> **Q7: On Line 219, the phrase "relationships between anchor points" is used twice in the same sentence. Is this a typographical error?**
>
> Yes, this section has been rewritten. Please see the updated manuscript.

---

### Official Review · Reviewer_A2GD · 2025-11-02

**Soundness:** 1
**Presentation:** 2
**Contribution:** 2
**Rating:** 4
**Confidence:** 5

**Summary:**

This paper proposed InertialGenome, a novel transformer-based framework for robust and resolution-agnostic chromosome reconstruction.  With this new design, it greatly improved the accuracy compraed to previous methods.

**Strengths:**

1. The background and problem setting is clearly addressed.
2. The network architecture is well designed and improved over previous graph-neural-network based approaches.
3. The performance improvement is impressive.

**Weaknesses:**

1. I think the problem setting has a siginificant limitation: there is no other ways to validate the predicted coordinates. In your setting, you compared the projected contact map from your 3D coordinates and the original contact map. However, single-cell Hi-C is very sparse and it can not fully capture the real 3D coordinates. Therefore, the reconstructed structure can be wrong even it has high agreement with the contact map.  Instead, I suggested to include chromatin tracing data to validate your methods, which you can refer to Higashi, which adopted some popular dataset.

2. The ablation study of the design is very limited, which I think need substantial improvement.
2.1 How to balance the structural stability loss and MSE loss? Are both losses needed?
2.2 What is the performances without inertial frame canonicalization?
2.3 What is ROPE's contribution to the performances?
2.4 What is the contribution of Nystr¨ om Approximation for Structure Tokenization?

3. The cross-resolution transfer benchmark is not so useful. We should expect we have different models for different resolutions, since they have very different biological focuses.

INERTIAL FRAME CANONICALIZATION

**Questions:**

1. Problem setting for real chromatin tracing data.
2. Ablation study to understand the key contribution of the designs.
3. The biological validation is not so meaningful, since you should expect to have similar TAD/compartment observations if you can make distance map close to the original distance/contact map, but that does not mean

---

> ### Author Response · Authors · 2025-11-22
> **Response**
>
> **Q1: suggest validating the results against imaging-based chromatin tracing data, similar to Higashi.**
>
> We thank the reviewer for this important point. We fully agree that contact map consistency alone is insufficient to validate 3D reconstructions from sparse single-cell Hi-C data. We explore the 4DN database and many experimental articles to find real experimental measurement validations that support our dataset.
>
> In response, we added FISH-based validation on GM12878 cells using spatial distances from Rao et al. [1]. As shown in Appendix H.3, our model correctly predicts that loop anchors (L1–L2) are closer than control pairs (L2–L3), consistent with FISH data and the inverse Hi-C contact trend—confirming biological plausibility beyond contact map fitting.
>
> We invite the reviewer to see Section H.3 for details.
>
> [1] Rao S S P, Huntley M H, Durand N C, et al. A 3D map of the human genome at kilobase resolution reveals principles of chromatin looping[J]. Cell, 2014, 159(7): 1665-1680.
>
> **Q2: The ablation study is very limited and needs substantial improvement.**
>
> We agree that a deeper study of individual design components can strengthen the technical contribution.
>
> **Q2.1. Balance between structural stability loss and MSE loss.**
>
> The overall training objective is a hybrid loss that combines a structural-preserving term and a value-weighted regression term:
>
> $L_{total} = \alpha L_{struct} + \beta L_{weighted\_mse}, \quad \beta = 1 - \alpha.$
>
> We performed sensitivity analysis by varying $\alpha/\beta$ in {0, 0.1, 0.5, 1.0}.
>
> | Ratio $\alpha/\beta$|320kb(dSCC$\uparrow$/dRMSE$\downarrow$)| 160kb(dSCC$\uparrow$/dRMSE$\downarrow$) | 80kb(dSCC$\uparrow$/dRMSE$\downarrow$) |40kb(dSCC$\uparrow$/dRMSE$\downarrow$)|
> | -------- | -------- | -------- |-------- |-------- |
> | 0.0 / 1.0  | 0.9030/0.1728 | 0.8627/0.1935 |0.7532/0.2152|0.7132/0.2410
> | 0.1 / 0.9  | **0.9029/0.1696** | **0.8595/0.1848** |**0.7663/0.2197**|**0.7158/0.2407**
> | 0.5 / 0.5  | **0.9002/0.1671** | **0.8580/0.1879** |**0.7741/0.2266**|**0.7203/0.2445**
> | 1.0 / 0.0  | 0.8815/0.1453 | 0.8484/0.1775 |0.7677/0.2297|0.7192/0.2788
>
> As shown in Table, removing structural supervision entirely ($\alpha = 0$) yields reasonable performance at coarse resolutions but leads to significant degradation in dRMSE at fine scales (e.g., 40kb), indicating poor geometric consistency. In contrast, using only structural loss ($\alpha = 1$) improves dRMSE at coarse resolutions but harms distance correlation (dSCC). A certain proportion of structural regularization ($\alpha = 0.1,0.5$) consistently achieves the best trade-off across all resolutions.
>
> Please see the updated manuscript (line 417-419 see Appendix G.1).
>
> **Q2.2. Without inertial frame canonicalization**
>
> **Q2.3. Contribution of ROPE**
>
> **Q2.4. Contribution of Nystrom Approximation**
>
> We conducte ablation experiments to address the Q2.2,2.3,2.4 mentioned above.
>
> | Model|320kb(dSCC$\uparrow$/dRMSE$\downarrow$)| 160kb(dSCC$\uparrow$/dRMSE$\downarrow$) | 80kb(dSCC$\uparrow$/dRMSE$\downarrow$) |40kb(dSCC$\uparrow$/dRMSE$\downarrow$)|
> | -------- | -------- | -------- |-------- |-------- |
> | Full (Ours)  | **0.9030/0.1547** | **0.8621/0.1809** |**0.7757/0.2035**|**0.7297/0.2382**
> | w/o Inertial | 0.9008/0.1641 | 0.8598/0.1869 |0.7737/0.2185|0.7226/0.2385
> | w/o RoPE     | 0.8976/0.1613 | 0.8566/0.1894 |0.7709/0.2229|0.7213/0.2454
> | w/o Nyström  | 0.9002/0.1659 | 0.8607/0.1998 |0.7746/0.2218|0.7214/0.2496
>
> We assess key components by ablating inertial alignment, RoPE-3D, or Nystr"om encoding in IG-3DMAX. Results show: (1) removing inertial alignment increases dRMSE across resolutions (e.g., 0.1547 → 0.1641 at 320 kb), confirming its role in global stability; (2) ablating RoPE degrades both dSCC and dRMSE, highlighting its importance for structural fidelity; and (3) disabling Nystr"om causes the largest drop at fine scales (e.g., +0.0114 dRMSE at 40 kb), underscoring its value in modeling long-range geometry. The full model consistently outperforms all variants.
>
> Please see the updated manuscript (line 420-421 see Appendix G.2).
>
> **Q3: The cross-resolution transfer benchmark is not so useful.**
>
> Thank you for the insightful comment. We agree that different resolutions capture distinct biological features and that specialized models can be useful. Our cross-resolution transfer experiment is not meant to replace resolution-specific modeling, but rather to explore whether low-resolution Hi-C—often more accessible—can guide accurate high-resolution 3D reconstruction by leveraging the hierarchical and geometrically consistent nature of chromatin. Results show our model successfully transfers structural priors across scales, improving fine-resolution accuracy (e.g., higher dSCC), which holds practical value when high-resolution data is scarce or costly.

---

### Author Response · Authors · 2025-11-27
**Latest Response**

## To All Reviewers

We sincerely thank all the reviewers for their detailed and constructive feedback. Several comments were particularly insightful and have prompted us to further deepen our understanding of the problem.

* A recurring concern from reviewers **A2GD**, **FLe5**, and **M6rw** is the need for stronger biological validation beyond Hi-C contact consistency. In response, we added FISH-based spatial validation on GM12878 cells using loop-anchor distances from Rao et al. (Cell, 2014). As shown in Appendix H.3, our model correctly predicts that loop anchors (L1–L2) are spatially closer than control pairs (L2–L3)—consistent with both FISH measurements and the inverse Hi-C contact trend—**demonstrating biological plausibility beyond mere contact fitting**.

* Multiple reviewers (**YNMs**, **FLe5**, **M6rw**) raised questions about methodological novelty, particularly regarding 3D-RoPE and its relation to recent work like STRING (ICML 2025). We clarify that while both extend rotary positional encoding to 3D, our 3D-RoPE is analytic, parameter-free, and chromatin-specific: it enforces distance-aware attention via fixed cosine–sine modulation across three orthogonal planes and is tightly integrated with inertial-frame canonicalization to respect geometric priors of chromatin folding. In contrast, STRING uses learnable skew-symmetric generators within a matrix exponential, making it data-driven. A concise comparison is provided in the revised manuscript (Appendix C).

* Reviewers **A2GD**, **YNMs**, and **M6rw** requested more comprehensive ablation studies. We have expanded our analysis across four resolutions (320 kb → 40 kb) to evaluate:
  – firstly, the balance between structural and regression losses (α/β trade-off),
  – secondly, the individual contributions of inertial frame canonicalization, 3D-RoPE, and Nyström approximation.
  Results (Appendix G) confirm that: (1) inertial alignment improves global geometric stability (↓dRMSE); (2) 3D-RoPE is critical for structural fidelity (↑dSCC, ↓dRMSE); and (3) Nyström approximation is especially valuable at fine scales for modeling long-range interactions. **The full model consistently outperforms all ablated variants**.

* Reviewer **M6rw** highlighted the absence of ChromFormer as a baseline. We implemented and evaluated it under identical conditions. However, ChromFormer designed for dense bulk Hi-C—relies on synthetic (Hi-C, 3D) pairs and becomes computationally prohibitive beyond 320 kb. At this resolution, our method achieves **0.9030 dSCC vs. 0.5478** for ChromFormer, underscoring that our gains stem not from the Transformer backbone alone, but from innovations tailored to chromosome reconstruction: **inertial canonicalization, geometry-aware encoding, and resolution-agnostic design**.

* Finally, reviewers **FLe5** and **YNMs** noted presentation issues (e.g., symbol definitions, vague phrasing, incomplete related work). We have thoroughly revised Sections 3.2 and the Introduction: clarified notation (e.g., $u_i$, $A_g$), added citations (Williams & Seeger for Nyström), distinguished genome-wide vs. targeted conformation methods, and sharpened our critique of prior work—particularly the over-constrained symmetry in E(3)-equivariant models like HiCEGNN, which impedes modeling of asymmetric features such as CTCF loops.

Our core contribution remains: **InertialGenome** is the first framework to combine inertial-frame canonicalization, analytic 3D positional encoding, and Nyström approximation for **robust, resolution-agnostic 3D chromosome reconstruction from sparse single-cell Hi-C data**—validated across multiple metrics, resolutions, and biological tasks.

We kindly ask the reviewers to consider our detailed responses and the substantial revisions made to the manuscript, and we would be grateful for your prompt provision of the latest feedback.

---

### Meta-Review · Area_Chair_szVh · 2026-01-07

**Summary:**

This paper proposes a resolution agnostic framework for 3D chromosome reconstruction from single cell images. The method leverages an inertial frame for pose canonicalization and proposes a geometry-aware positional encoding for a transformer-based reconstruction model. Experimental results demonstrate that the proposed method provides biologically plausible reconstructions and achieves improved performance.

The initial reviewer evaluation of this paper leans toward rejection, with scores of Reject (2), Marginally Below Acceptance Threshold (4), and Marginally Above Acceptance Threshold (6). The ratings are somewhat divergent.

The reviewers raised several major concerns:

1. Limited problem setting, with insufficient experimental validation beyond contact consistency.
2. Need for ablation studies and more in-depth analysis.
3. Motivation and merit of resolution-agnostic models, including the absence of cross-resolution transfer benchmarks.
4. Questions regarding the novelty of the proposed method, including RoPE-3D.
5. Issues related to writing clarity and the coverage of related work.

**Reviewer Concerns:**

The authors have extensively addressed the issues raised by the reviewers, including requests for ablation studies and suggestions on presentation.

1. Reviewer A2GD pointed out that contact map consistency alone is not sufficient to validate 3D reconstruction and requested validation using chromatin tracing data, as in Higashi. The authors responded with an in-depth analysis using HiCCUPS, presented in Section H.3 of the appendix.
2. An ablation study was provided in the rebuttal, addressing concerns regarding the contribution of individual components.
3. This point is partially addressed. The authors acknowledged that resolution-specific models remain more powerful, and that the practicality of the proposed method is limited. However, improvement at finer resolutions indicates that the proposed method positively transfer priors across scales.
4. The authors clarified the novelty of 3D-RoPE by discussing key distinctions between the proposed RoPE and prior work, including the fact that the proposed positional encoding is chromatin-specific and prior-driven, which differs considerably from existing approaches such as STRING.
5. All writing-related suggestions were incorporated into the revised manuscript, including the addition of the recommended related work.

**Reviewer Scores:**

Despite the mixed and negative initial scores, the rebuttal has considerably strengthened the paper. The authors have addressed the issues, including more rigorous validation of 3D reconstruction and ablation studies supported by additional experimental results. However, given the inferior performance of resolution-agnostic models compared to resolution-specific models, the merit of this work remains limited. In conclusion, if a full discussion period given, the overall ratings would likely increase. Nevertheless, due to the negative initial ratings and the limited practicality and generality of the proposed methods, I recommend acceptance with reservations.

---

### Decision · Program_Chairs · 2026-01-26

Accept (Poster)